# Feature Unlearning: Theoretical Foundations and Practical Applications with Shuffling

**Yue Yang**[*]
yue@maincode.com
Maincode; Monash University

**Jinhao Li**[*]
jinhao.li@monash.edu
Monash University

**Hao Wang**[†]
hao.wang2@monash.edu
Monash University

## Abstract

Machine unlearning has become a focal point in recent research, yet the specific area of feature unlearning has not been thoroughly explored. Feature unlearning involves eliminating specific features' effects from an already trained model, presenting distinct challenges that are not yet comprehensively addressed. This paper presents a novel and straightforward approach to feature unlearning that employs a tactical shuffling of the features designated for removal. By redistributing the values of the features targeted for unlearning throughout the original training dataset and subsequently fine-tuning the model with this shuffled data, our proposed method provides a theoretical guarantee for effective feature unlearning. Under mild assumptions, our method can effectively disrupt the established correlations between unlearned features and the label, while preserving the relationships between the remaining features and the label. Across both tabular and image datasets, our empirical results show that our method not only effectively and efficiently removes the influence of designated features but also preserves the information content of the remaining features.

## 1 Introduction

Machine learning models have transformed numerous industries by learning complex patterns from vast amounts of data, driving productivity and efficiency [29, 19, 24]. However, as these models become deeply integrated into decision-making processes, significant concerns regarding privacy, ethical use of data, and compliance with regulatory frameworks, such as the General Data Protection Regulation [31] and the Health Insurance Portability and Accountability Act [1], have surged. In this context, machine unlearning has emerged as a critical subfield, focusing on the deliberate removal of specific information from trained models, effectively enabling them to "forget" certain data points or features upon request [27, 16, 5, 14]. While machine unlearning has garnered significant attention, the specific subfield of feature-level unlearning remains relatively underexplored.

Most existing research on machine unlearning has focused on instance-based unlearning, aiming to remove a certain subset of data samples from models without retraining from scratch [8, 27, 14, 16]. In contrast, feature unlearning targets removing the influence of specific features from a trained model [32], which poses unique challenges and opportunities that have yet to be fully addressed. One of the major challenges for feature unlearning lies in the complex interdependencies among features: disentangling and removing the effect of a targeted feature without disrupting the influence of others is difficult. The model may have learned complex representations where the influence of one feature is intertwined with others, making isolation of a single feature's effect non-trivial. Moreover, changes to one feature's influence can inadvertently affect the learned relationships and decision boundaries associated with other features, potentially leading to unintended bias.

---

[*]Equal contribution.
[†]Corresponding author

39th Conference on Neural Information Processing Systems (NeurIPS 2025).

Existing approaches for feature-level unlearning [32, 26, 15] experience performance degradation as the number of removed features increases, and they lack theoretical guarantees to fully negate the influence of removed features while maintaining the integrity of the remaining features.

To tackle this issue, in this paper, as shown in Fig. 1, we introduce a simple yet effective approach to feature unlearning by strategically shuffling the features designated for removal from a statistical perspective. This shuffling technique serves a dual purpose. First, it ensures that the modified model gradually loses reliance on the specified features, as the original correlations are obscured through randomization. Second, this approach preserves the structure and statistical properties of the remaining features, thereby maintaining the integrity of other predictive relationships in the data.

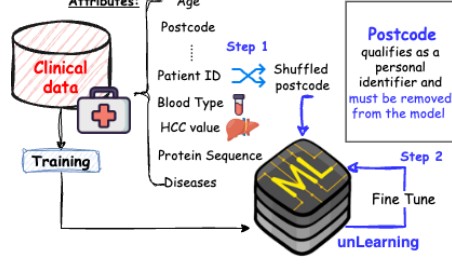

Figure 1: An example of our feature unlearning algorithm.

Despite its simplicity, we provide rigorous theoretical guarantees for the feature unlearning process. We demonstrate that the impact of the specified features diminishes to an insignificant level. We also further validate our proposed method under the concept of Shapley value [33] and mutual information [20]. These guarantees provide assurance that the shuffling process effectively isolates and removes the targeted feature influence without compromising overall model integrity. Through comprehensive empirical evaluation on diverse tabular and image datasets with different neural architectures, we show that our approach can robustly unlearn multiple features simultaneously while efficiently preserving high accuracy and ensuring strong generalization on the retained features.

## 2 Related Work

Machine unlearning has emerged as a critical area in machine learning, focusing on enabling models to efficiently forget specific data points or subsets without requiring a complete retraining [27, 16, 5, 14, 13, 3]. Most existing machine unlearning methods primarily focused on data-level unlearning and proposed several methods, such as Sharded, Isolated, Sliced, and Aggregated (SISA) training, certified removal, and adaptive unlearning, which can effectively remove specific data points [5, 14, 16].

There have been comparatively limited studies [32, 15, 26] addressing feature-level unlearning, which aims to remove or diminish the influence of specific features in a trained model. [32] initially addressed the feature unlearning problem by recalculating the network weights associated with the removed elements, followed by a fine-tuning phase to maintain performance on remaining features and labels. Subsequently, [15] proposed a more computationally efficient approach to eliminate specific attributes using a representation detachment loss defined by mutual information. Yet, this method lacks a theoretical guarantee for the unlearning process. Also, mutual information is simultaneously estimated during unlearning, introducing potential bias. [26] expanded feature unlearning to pre-trained generative models by modifying the generator and encoder-decoder pathways. However, this method is specifically tailored for the generative adversarial network framework and can lead to considerable computational overhead. All of the aforementioned models suffer from one of the two main limitations: (i) Their performance deteriorates as the number of features to be removed increases; (ii) They lack theoretical guarantees to completely eliminate the influence of removed features while preserving the information of the remaining features. Instead of simply retraining the model from scratch, prior studies often prefer fine-tuning a trained model because it only need to work on a part of the parameter space that already captures wide-ranging and sturdy features and make minimal yet effective adjustments rather than relearning all features from the beginning [35, 10]. Also, beginning with a pre-trained model acts as a form of prior, effectively limiting the model's degrees of freedom. This constraint makes it less prone to fitting random noise and more likely to preserve features that generalize well. Consequently, this kind of approach tends to reduce overfitting compared to training a model from scratch [23].

## 3 Defining Feature Unlearning

**Definition 3.1** (*Feature Unlearning*). Let $\hat{\mathcal{D}} = \{(\mathbf{x}_i, y_i)\}_{i=1}^n$ be a dataset of $n$ samples with $m$ features, where the feature vector $\mathbf{x_i} \in \mathcal{X} \subseteq \mathbb{R}^m$ and the label $y_i \in \mathcal{Y} \subseteq \mathbb{R}$. Each feature, as well

as the label, can be represented as a random variable, denoted by $X_1, X_2, \ldots, X_m, Y$. The random variable of the feature vector is then denoted by $\mathbf{X} = (X_1, \ldots, X_m)$. Each sample in the dataset $\hat{\mathcal{D}}$ is drawn from its underlying distribution $\mathcal{D}$, where $(\mathbf{X}, Y) \sim \mathcal{D}$.

Suppose a model parameterized by $\theta$, denoted by $f_\theta : \mathcal{X} \to \mathcal{Y}$, has been trained on $\hat{\mathcal{D}}$. Let the $j$-th feature (which could also include multiple features as a feature mix) be the targeted feature for unlearning. The dataset with $j$-th feature removed is defined as $\hat{\mathcal{D}}_{-j} := \{(\mathbf{x}_{i,-j}, y_i)\}_{i=1}^n$, where the resulting feature vector $\mathbf{x}_{i,-j} \in \mathcal{X}_{-j} \subseteq \mathbb{R}^{m-1}$. Let $g_\phi : \mathcal{X}_{-j} \to \mathcal{Y}$ be the model parameterized by $\phi$ and trained from *scratch* on $\hat{\mathcal{D}}_{-j}$.

We define the *unlearning operator* $\mathcal{U}$ transforming the original model into an updated model, expressed as $f_{\theta'} = \mathcal{U}(f_\theta)$, such that the influence of the $j$-th feature is completely removed. We define that $\mathcal{U}$ achieves the goal of *feature unlearning* if, with a probability of at least $1 - \delta$, for all $(\mathbf{x}, y)$ drawn from $\hat{\mathcal{D}}$, there exists a $g_\phi$, such that $\Delta\big(f_{\theta'}(\mathbf{x}), g_\phi(\mathbf{x}_{-j})\big) \leq \epsilon, \quad \forall (\mathbf{x}, y) \sim \hat{\mathcal{D}}$, where $\Delta(\cdot, \cdot)$ is an appropriate distance function. In other words, we have $P_{(\mathbf{x}, y) \sim \hat{\mathcal{D}}}\Big[\Delta\big(f_{\theta'}(\mathbf{x}), g_\phi(\mathbf{x}_{-j})\big) > \epsilon\Big] \leq \delta$.

# 4   Shuffling-based Feature Unlearning Approach

In the context of *feature unlearning*, our objective is to ensure that the updated model $f_{\theta'}$ retains no knowledge of the specific feature to be unlearned. Formally, this implies that $f_{\theta'}$ should not encode any meaningful information about the values of the removed feature, i.e., the model's parameters are *independent* of the unlearned feature. Meanwhile, it is crucial that the unlearning process does not degrade information pertaining to other features. In other words, $f_{\theta'}$ should continue to capture the useful correlations present in the remaining dataset excluding the removed feature. In this section, we first introduce a straightforward yet highly effective method for unlearning a feature from the dataset, followed by a theoretical analysis to establish a solid foundation for our method.

**Approach Overview:** Our proposed feature unlearning consists of two key steps: *random shuffling* and *fine-tuning*. In the first step, the $j$-th feature is shuffled while its marginal probability distribution remains unchanged. Meanwhile, as theoretically demonstrated in our work, this shuffling process effectively disrupts any alignment between $X_j$ and $(\mathbf{X}_{-j}, Y)$, i.e., making $X_j^\pi$ and $(\mathbf{X}_{-j}, Y)$ independent. The second step involves training the original model using the same loss function but with the shuffled dataset. The resulted model theoretically loses its ability to exploit the original correlation between feature $X_j$ and the target $Y$, while retaining the information associated with remaining features $\mathbf{X}_{-j}$. Consequently, $f_{\theta'}$ has "forgotten" the specific predictive information provided by feature $X_j$ in the original dataset $\hat{\mathcal{D}}$.

### Feature Random Shuffling

i Draw a random permutation $\pi$ of the index set $\{1, \ldots, n\}$ uniformly at random.

ii Construct the *shuffled dataset* $\hat{\mathcal{D}}_j^\pi$ by replacing each $x_{i,j}$ with $x_{\pi(i),j}$ while keeping the value of other features the same for each data point. The shuffled dataset is defined as $\hat{\mathcal{D}}_j^\pi := \{(\mathbf{x}_i^\pi, y_i)\}_{i=1}^n$, where $\mathbf{x}_i^\pi = (x_{i,1}, \ldots, x_{i,j-1}, x_{\pi(i),j}, x_{i,j+1}, \ldots, x_{i,m})$. That is, all features *other* than $j$ remain unchanged, while the entries of the $j$-th feature are permuted.

**Fine-Tuning with the Original Loss**   Suppose we have a loss function, defined as $\ell(f_\theta(\mathbf{x}), y)$, that was used to train the original model. In our approach, we fine-tune the original model using the same loss and the shuffled dataset $\hat{\mathcal{D}}_j^\pi$ to get the updated parameters $\theta'$.

## 4.1   Theoretical Analysis

**Independency Between $X_j^\pi$ and $(\mathbf{X}_{-j}, Y)$**   We assume the distribution $\mathcal{D}$ is stationary and ergodic on the measurable space of $\mathcal{X} \times \mathcal{Y}$, implying that each coordinate $x_{i,j}$ and label $y_i$ also forms a stationary and ergodic process, denoted by $\{x_{i,j}, y_i\}_{i=1}^n$, in the smaller space $\mathcal{X}_j \times \mathcal{Y}$. The shuffled sequence through the random permutation $\pi$ is written as $\{x_{\pi(i),j}, y_i\}_{i=1}^n$. We first define the empirical distribution of the shuffled sequence as $\hat{P}_{(X_j^\pi, Y)}^{(n)} := \frac{1}{n} \sum_{i=1}^n \delta(x_{\pi(i),j}, y_i)$, where $\delta(x_{\pi(i),j}, y_i)$ is a Dirac point mass at $(x_{\pi(i),j}, y_i)$.

Here, we first prove the independency between $X_j^\pi$ and $(\mathbf{X}_{-j}, Y)$ after random shuffling.

**Theorem 4.1.** *Over the random permutation $\pi$ and the sampled datasets $\hat{\mathcal{D}}$ and $\hat{\mathcal{D}}_j^\pi$,*

$$\hat{P}_{(X_j^\pi, Y)}^{(n)} \xrightarrow[n\to\infty]{\text{a.s.}} P_{X_j} \times P_Y, \qquad \hat{P}_{(X_j^\pi, \mathbf{X}_{-j})}^{(n)} \xrightarrow[n\to\infty]{\text{a.s.}} P_{X_j} \times P_{\mathbf{X}_{-j}}. \tag{1}$$

*Proof.* In this proof, we provide a comprehensive demonstration for the independency between $X_j$ and $Y$, which is analogous to that of the independency between $X_j$ and $\mathbf{X}_{-j}$ with full details presented in Appendix A.

As $\{x_{i,j}, y_i\}_{i=1}^n$ is stationary and ergodic, the coordinate processes $\{x_{i,j}\}_{i=1}^n$ and $\{y_i\}_{i=1}^n$ both satisfy the *Birkhoff's ergodic theorem* [4], which means that, for each measurable $A \subseteq \mathcal{X}_j$ and $B \subseteq \mathcal{Y}$:

$$\hat{P}_{X_j}^{(n)}(A) = \frac{1}{n}\sum_{i=1}^n \mathbf{1}\{x_{i,j} \in A\} \xrightarrow[n\to\infty]{\text{a.s.}} P_{X_j}(A), \; \hat{P}_Y^{(n)}(B) = \frac{1}{n}\sum_{i=1}^n \mathbf{1}\{y_i \in B\} \xrightarrow[n\to\infty]{\text{a.s.}} P_Y(B). \tag{2}$$

Thus, $\hat{P}_{X_j}^{(n)} \to P_{X_j}$ and $\hat{P}_Y^{(n)} \to P_Y$ almost surely.

The shuffled empirical distribution of a rectangle $A \times B$ is written as

$$\hat{P}_{X_j^\pi, Y}^{(n)}(A \times B) = \frac{1}{n}\sum_{i=1}^n \mathbf{1}\{x_{\pi(i),j} \in A, y_i \in B\}. \tag{3}$$

For this empirical distribution, when conditioning on the process $\{x_{i,j}, y_i\}_{i=1}^n$, for each $i$, the index $\pi(i)$ is uniform in $\{1, \ldots, n\}$. Therefore, we have

$$\mathbb{E}_\pi[\mathbf{1}\{x_{\pi(i),j} \in A\}|\{x_{i,j}, y_i\}_{i=1}^n] = \hat{P}_{x_j}^{(n)}(A). \tag{4}$$

Multiplying by $\mathbf{1}\{y_i \in B\}$ and summing over $i$ then shows

$$\mathbb{E}_\pi[\hat{P}_{X_j^\pi, Y}^{(n)}(A \times B)|\{x_{i,j}, y_i\}_{i=1}^n] = \hat{P}_{X_j}^{(n)}(A)\hat{P}_Y^{(n)}(B). \tag{5}$$

We then un-condition (remove the conditioning of) the above expectation using the *law of total expectation* as

$$\mathbb{E}_\pi[\hat{P}_{X_j^\pi, Y_i}^{(n)}(A \times B)] = \mathbb{E}_{\{x_{i,j}, y_i\}_{i=1}^n}[\hat{P}_{X_j}^{(n)}(A)\hat{P}_Y^{(n)}(B)] = \hat{P}_{X_j}^{(n)}(A)\hat{P}_Y^{(n)}(B). \tag{6}$$

Notably, the last term is derived based on the fact that the product of two marginal probabilities is constant when fixing $(x_{\pi(i),j}, y_i)$, i.e., there is no more randomness in $\pi$ for the marginals.

Given Eq. (6), by defining the difference between the shuffled empirical distributions and the product of two marginal empirical distributions, i.e.,

$$\Omega_n(A, B) := \hat{P}_{X_j^\pi, Y}^{(n)}(A \times B) - \hat{P}_{X_j}^{(n)}(A)\hat{P}_Y^{(n)}(B). \tag{7}$$

We see that $\mathbb{E}_\pi[\Omega_n(A, B)] = 0$, i.e., $\Omega_n(A, B)$ is mean zero.

Next, we aim to show that $\Omega_n(A, B)$ is *small with a high probability*. To prove this we first define $S_n(A, B) = \sum_{i=1}^n \mathbf{1}\{x_{\pi(i),j} \in A, y_i \in B\}$. Then, the shuffled empirical distribution $\hat{P}_{X_j^\pi, Y}^{(n)}(A \times B)$ can be expressed as $\frac{1}{n}S_n(A, B)$. Changing $\pi$ in one position affects at most *two* indicators in $S_n$. Hence, the rewritten shuffled empirical distribution $\frac{1}{n}S_n(A, B)$ satisfies the *McDiarmid's inequality* [25] – for some $c_i > 0$ as per-sample sensitivity constants and $\epsilon > 0$,

$$P(|\frac{1}{n}S_n(A, B) - \mathbb{E}_\pi[\frac{1}{n}S_n(A, B)]| > \epsilon) \leq 2\exp\left(-\frac{2\varepsilon^2}{\sum_{i=1}^n c_i^2}\right). \tag{8}$$

Given Eq. (6), we have $\mathbb{E}_\pi[\frac{1}{n}S_n(A, B)] = \hat{P}_{X_j}^{(n)}(A)\hat{P}_Y^{(n)}(B)$. The McDiarmid's inequality in Eq. (8) can be rewritten as

$$P(|\hat{P}_{X_j^\pi, Y}^{(n)}(A \times B) - \hat{P}_{X_j}^{(n)}(A)\hat{P}_Y^{(n)}(B)| > \epsilon) = P(|\Omega_n(A, B)| > \epsilon) \leq 2\exp\left(-\frac{2\varepsilon^2}{\sum_{i=1}^n c_i^2}\right). \tag{9}$$

Then, summing over $n$, we have $\sum_{n=1}^{\infty} P(|\Omega_n(A, B)| > \epsilon) \leq \infty$, which, by the *Borel-Cantelli* lemma [36, 7], results in $\Omega_n(A, B) \to 0$ almost surely for each fixed $A \times B$. Combined with the *Birkhoff's ergodic theorem* in Eq. (2) (i.e., the empirical marginal distribution almost surely converges to its marginal distribution), we have

$$\hat{P}_{X_j^\pi, Y}^{(n)}(A \times B) = \hat{P}_{X_j}^{(n)}(A)\hat{P}_Y^{(n)}(B) + \Omega_n(A, B) \xrightarrow[n\to\infty]{\text{a.s.}} P_{X_j}(A) \times P_Y(B). \qquad (10)$$

So far, we have *point-wise* almost sure convergence for each rectangle $A \times B$. To further show the shuffled empirical distribution $\hat{P}_{X_j^\pi, Y}^{(n)}(A \times B)$ almost surely converges to $P_{X_j} \times P_Y$ on *all* measurable sets, we choose a countable family of rectangles $\{A_l \times B_l\}$ that generates the product $\sigma$-algebra on $\mathcal{X}_j \times \mathcal{Y}$. By the same exponential tail bound in Eq. (9), a union of the Borel-Cantelli lemma shows $|\Omega_n(A_l, B_l)| \to 0$ simultaneously for all $l$ with the probability of 1.

Since these rectangles form (or generate) a $\pi$-system for $\mathcal{X}_j \times \mathcal{Y}$, the standard measure-theoretic uniqueness result implies $\hat{P}_{X_j^\pi, Y}^{(n)} \xrightarrow[n\to\infty]{\text{a.s.}} P_{X_j} \times P_Y$. $\qquad \square$

**Corollary 4.2.** *If $\hat{P}_{X_j^\pi, Y}^{(n)} \xrightarrow[n\to\infty]{\text{a.s.}} P_{X_j} \times P_Y$ and $\hat{P}_{X_j^\pi, \mathbf{X}_{-j}}^{(n)} \xrightarrow[n\to\infty]{\text{a.s.}} P_{X_j} \times P_{\mathbf{X}_{-j}}$, then the shuffled feature $X_j^\pi$ is almost surely independent of both the label $Y$ and the unshuffled features $\mathbf{X}_{-j}$, such that $X_j^\pi \perp\!\!\!\perp (Y, \mathbf{X}_{-j})$.*

*Proof.* Refer to Appendix B. $\qquad \square$

*Remark* 4.3. *The above corollary aligns with our previous statement: our method effectively disrupts the established correlations between unlearned features and the label while preserving the relationships between the remaining features and the label. The former is evident, as the shuffled feature becomes independent of the label. The latter is also satisfied since our approach does not modify either the remaining features or the label, thereby maintaining their original relationship.*

**Proof of $(\epsilon - \delta)$-Close Between $f_{\theta'}$ and $g_\phi$**

**Theorem 4.4.** *If $X_j^\pi$ is independent of $(Y, \mathbf{X}_{-j})$, then using the same loss function $\ell : \mathcal{Y} \times \mathcal{Y} \to \mathbb{R}_+$ and the same optimizer, the model $f_{\theta'}$ trained on the shuffled dataset $\hat{\mathcal{D}}_j^\pi$ and the model $g_\phi$ trained from scratch on $\hat{\mathcal{D}}_{-j}$ are $(\epsilon, \delta)$-close.*

*Proof.* Let $\mathcal{F}$ be the hypothesis class for functions $f(\mathbf{x})$ with domain $\mathcal{X} \subseteq \mathbb{R}^m$ and $\mathcal{G}$ be the hypothesis class for functions $g(\mathbf{x}_{-j})$ with domain $\mathcal{X}_{-j} \subseteq \mathbb{R}^{m-1}$.

We define the *optimal* risk in the classes of $\mathcal{F}$ and $\mathcal{G}$ as

$$\mathcal{R}^* = \inf_{f \in \mathcal{F}} \left\{ \mathbb{E}_{(\mathbf{x}^\pi, y) \sim \mathcal{D}_j^\pi} \left[ \ell\left(f(\mathbf{x}^\pi), y\right) \right] \right\}, \mathcal{R}_{-j}^* = \inf_{g \in \mathcal{G}} \left\{ \mathbb{E}_{(\mathbf{x}_{-j}, y) \sim \mathcal{D}_{-j}} \left[ \ell\left(g(\mathbf{x}_{-j}), y\right) \right] \right\}. \quad (11)$$

Since $X_j^\pi$ is independent of $(Y, \mathbf{X}_{-j})$, the conditional distribution of $y$ given $(\mathbf{x}_{-j}, x_j^\pi)$ equals that of $y$ given $\mathbf{x}_{-j}$ alone. Therefore, a function that tries to exploit $x_j^\pi$ cannot improve its predictions beyond what is already possible with $\mathbf{x}_{-j}$ alone. Specifically, for any $f(\mathbf{x}) \in \mathcal{F}$, we define $\tilde{g}(\mathbf{x}_{-j}) := f(\mathbf{x}_{-j}, x_j^\pi)$ for an arbitrary fixed value of $x_j^\pi$. The expected loss of $\tilde{g}$ under $\hat{\mathcal{D}}_{-j}$ matches that of $f$ under $\mathcal{D}_j^\pi$. Therefore, we have

$$\inf_{f \in \mathcal{F}} \left\{ \mathbb{E}_{\mathcal{D}_j^\pi} \left[ \ell\left(f(\mathbf{x}^\pi), y\right) \right] \right\} = \inf_{\tilde{g} \in \mathcal{G}} \left\{ \mathbb{E}_{\mathcal{D}_{-j}} \left[ \ell\left(\tilde{g}(\mathbf{x}_{-j}), y\right) \right] \right\} = \mathcal{R}_{-j}^*. \quad (12)$$

We assume that the model $f_{\theta'}$ trained on the shuffled dataset $\hat{\mathcal{D}}_j^\pi$ and the model $g_\phi$ trained on the dataset $\hat{\mathcal{D}}_{-j}$ aim to minimize their respective empirical risks. That is, each corresponds to a solution (or approximate solution) of empirical risk minimization (ERM), such that

$$\hat{\mathcal{R}}_{\hat{\mathcal{D}}_j^\pi}(f_{\theta'}) \leq \inf_{f \in \mathcal{F}} \hat{\mathcal{R}}_{\hat{\mathcal{D}}_j^\pi}(f) + \alpha, \qquad \hat{\mathcal{R}}_{\hat{\mathcal{D}}_{-j}}(g_\phi) \leq \inf_{g \in \mathcal{G}} \hat{\mathcal{R}}_{\hat{\mathcal{D}}_{-j}}(g) + \alpha, \quad (13)$$

where $\alpha > 0$ is the small sub-optimality that may come from incomplete training or early stopping.

By the standard uniform convergence theorem, with a probability of at least $1 - \delta$ over the draw of $\hat{\mathcal{D}}$ and any randomness in forming $\hat{\mathcal{D}}_j^\pi$ or in the training algorithm, we simultaneously have

$$|\hat{R}_{\hat{\mathcal{D}}_j^\pi}(f) - \mathcal{R}(f)| \le \epsilon_n, \quad \forall f \in \mathcal{F}, \qquad |\hat{R}_{\hat{\mathcal{D}}_{-j}}(g) - \mathcal{R}(g)| \le \epsilon_n, \quad \forall g \in \mathcal{G}, \qquad (14)$$

for some bound $\epsilon_n$ that goes to 0 as $n \to \infty$, depending on the complexities of $\mathcal{F}$ and $\mathcal{G}$.

From the above uniform convergence bound, we have

$$\mathcal{R}(f_{\theta'}) \le \hat{\mathcal{R}}_{\hat{\mathcal{D}}_j^\pi}(f_{\theta'}) + \epsilon_n \le \inf_{f \in \mathcal{F}} \hat{\mathcal{R}}_{\hat{\mathcal{D}}_j^\pi}(f) + \alpha + \epsilon_n \le \inf_{f \in \mathcal{F}} \mathcal{R}(f) + 2\epsilon_n + \alpha = R^* + 2\epsilon_n + \alpha. \quad (15)$$

Similarly, we have $\mathcal{R}(g_\phi) \le \mathcal{R}_{-j}^* + 2\epsilon_n + \alpha$. By the triangle inequality, we can further derive the following inequalities for risks and empirical risks with the probability of at least $1 - \delta$ as

$$|\mathcal{R}(f_{\theta'}) - \mathcal{R}(g_\phi)| \le |\mathcal{R}(f_{\theta'}) - \mathcal{R}^*| + |\mathcal{R}(g_\phi) - \mathcal{R}^*| \le 4\epsilon_n + 2\alpha, \qquad (16)$$

and

$$|\hat{\mathcal{R}}_{\hat{\mathcal{D}}_j^\pi}(f_{\theta'}) - \hat{\mathcal{R}}_{\hat{\mathcal{D}}_{-j}}(g_\phi)| \le 6\epsilon_n + 2\alpha. \qquad (17)$$

Therefore, the empirical risk difference between $f_{\theta'}$ and $g_\phi$ is $(\epsilon - \delta)$-close, where $\epsilon = 6\epsilon_n + 2\alpha$.

In standard supervised learning theory, the *Bayes-optimal* predictors (such as $f_{\theta'}$ and $g_\phi$) for $\ell$-based risk are given by

$$f_{\theta'} = \operatorname*{argmin}_{\theta'} \hat{\mathcal{R}}_{\hat{\mathcal{D}}_j^\pi}(f_{\theta'}), \qquad g_\phi = \operatorname*{argmin}_{\phi} \hat{\mathcal{R}}_{\hat{\mathcal{D}}_{-j}}(g_\phi). \qquad (18)$$

In the following, we will prove that: 1) if $\ell$ is $k$-strongly convex, for any $f_{\theta'}$ and $g_\phi$ trained via Eq. (18), they are $(\epsilon - \delta)$-close; 2) if $\ell$ is not strongly convex, under mild assumptions, for any $f_{\theta'}$ trained via Eq. (18), there must exist a $g_\phi$ trained via Eq. (18) such that $f_{\theta'}$ and $g_\phi$ are $(\epsilon - \delta)$-close.

The proof of the former is shown below and that of the latter is presented in Appendix C.

As $\ell$ is $k$-strongly convex, the ERM problem has a unique global minimizer, which means that $f_{\theta'}$ and $g_\phi$ obtained via Eq. (18) are unique. Also, at the minimizer, both gradients of the empirical risk are zero, i.e., $\nabla \hat{\mathcal{R}}_{\hat{\mathcal{D}}_j^\pi}(f_{\theta'}) = \nabla \hat{\mathcal{R}}_{\hat{\mathcal{D}}_{-j}}(g_\phi) = 0$.

Moreover, for any $f_{\theta'}$ and $g_\phi$ in their respective hypothesis classes, the strong convexity yields

$$\hat{\mathcal{R}}_{\hat{\mathcal{D}}_{-j}}(g_\phi) \ge \hat{\mathcal{R}}_{\hat{\mathcal{D}}_j^\pi}(f_{\theta'}) + \nabla \hat{\mathcal{R}}_{\hat{\mathcal{D}}_j^\pi}(f_{\theta'})^T (g_\phi - f_{\theta'}) + \frac{k}{2}||g_\phi - f_{\theta'}||^2 \ge \hat{\mathcal{R}}_{\hat{\mathcal{D}}_j^\pi}(f_{\theta'}) + \frac{k}{2}||g_\phi - f_{\theta'}||^2. \qquad (19)$$

By rearranging Eq. (19), we have

$$||g_\phi - f_{\theta'}||^2 \le \frac{2}{k}|\hat{\mathcal{R}}_{\hat{\mathcal{D}}_{-j}}(g_\phi) - \hat{\mathcal{R}}_{\hat{\mathcal{D}}_j^\pi}(f_{\theta'})| \le \frac{2}{k}(6\epsilon_n + 2\alpha). \qquad (20)$$

Therefore, the difference in risk of two models is bounded by $\sqrt{\frac{4}{k}(3\epsilon_n + \alpha)}$, which implies that $f_{\theta'}$ and $g_\phi$ are $(\epsilon - \delta)$-close, where $\epsilon = \sqrt{\frac{4}{k}(3\epsilon_n + \alpha)}$. $\qquad \square$

*Remark* 4.5 (*Insights from Mutual Information and Shapley Value*). *The unlearning capability of our shuffle-based approach can also be explained from the mutual information and the Shapley value perspective. For the former, the mutual information between the unlearned feature and the label vanishes, which is formally stated in the following Theorem 4.6. For the latter, the Shapley value of the unchanged features (i.e., $\mathbf{x}_{-j}$) derived from the unlearned model $f_{\theta'}$ and the retrained model $g_\phi$ are almost surely identical which is stated in Theorem 4.7.*

**Theorem 4.6.** *If $X_j^\pi \perp\!\!\!\perp (Y, \mathbf{X}_{-j})$ is almost sure when $n \to \infty$, then the empirical mutual information $I_n(X_j^\pi, Y) \xrightarrow[n \to \infty]{a.s.} 0$, $I_n(X_j^\pi, \mathbf{X}_{-j}) \xrightarrow[n \to \infty]{a.s.} 0$, and $I_n(X_j^\pi, \theta') \xrightarrow[n \to \infty]{a.s.} 0$.*

*Proof.* Refer to Appendix D. $\qquad \square$

**Theorem 4.7.** *Let $\kappa_i(f_{\theta'})$ and $\kappa_i(g_\phi)$ be the Shapley value of the $i$-th feature derived from models $f_{\theta'}$ and $g_\phi$. If $X_j^\pi \perp\!\!\!\perp (Y, \mathbf{X}_{-j})$ is almost sure when $n \to \infty$, $\forall i \in \{1, \ldots, j-1, j+1, \ldots, m\}$, we almost surely have $\kappa_i(f_{\theta'}) = \kappa_i(g_\phi)$.*

*Proof.* Refer to Appendix E. $\qquad \square$

# 5 Experimental Results

To align with the definition of feature unlearning presented in Definition 3.1, our experiments involve the following setup. For a given dataset $\hat{\mathcal{D}}$ and its corresponding model $f_\theta$, which has been fully trained on this dataset, the $j$-th feature in the training set (denoted as $\hat{\mathcal{D}}_{\text{Train}}$) is randomly shuffled to train $f_{\theta'}$ for unlearning. For the retrain from scratch model, the $j$-th feature is removed from $\hat{\mathcal{D}}_{\text{Train}}$ to train $g_\phi$. The test set, denoted as $\hat{\mathcal{D}}_{\text{Test}}$, is derived from $\hat{\mathcal{D}}$ using an 80:20 training-test split ratio. Numerical features undergo standardization, and categorical features are processed via one-hot encoding.

In this section, we first introduce several metrics to evaluate the effectiveness of feature unlearning in Section 5.1. We compare our proposed method's performance against a model retrained from scratch using the TRI metric and evaluated efficiency via the EI index. We also perform dependence analysis and assessed feature contributions as part of our evaluation framework. Subsequently, we present the experimental settings in Section 5.2, followed by the results for both single-feature and multi-feature unlearning in Sections 5.2.1 and 5.2.2, respectively. ***The codes are available in the supplementary materials.***

## 5.1 Evaluation Metrics

***TRI*** [6]: The *test retention index* (TRI) evaluates the effectiveness of feature unlearning by **comparing the accuracy of the unlearned model (i.e., $f_{\theta'}$) to a model trained from scratch (i.e., $g_\phi$)**, which is widely recognized as a gold standard to evaluate unlearning. TRI is defined as $TRI = \frac{Acc_{f_{\theta'}}}{Acc_{g_\phi}}$, where $Acc_{f_{\theta'}}$ and $Acc_{g_\phi}$ represent the test accuracy on $\hat{\mathcal{D}}_{\text{Test}}$ for $f_{\theta'}$ and $g_\phi$, respectively. A TRI value closer to one indicates effective feature unlearning.

***EI*** [6]: In addition to accuracy, to assess the **efficiency** of the unlearning process, we introduce the *efficiency index* (EI), defined as $EI = \frac{Time_{g_\phi}}{Time_{f_{\theta'}}}$, where $Time_{f_{\theta'}}$ and $Time_{g_\phi}$ denote the training times (in seconds) for $f_{\theta'}$ and $g_\phi$, respectively. An EI value greater than one suggests the unlearning algorithm is more time-efficient than training from scratch.

***RASI***: The *unlearning robustness against shuffling index* (RASI) examines whether the unlearned model has successfully eliminated **dependency** on the unlearned feature. This is evaluated by shuffling the values of the unlearned feature in $\hat{\mathcal{D}}_{\text{Test}}$ using the `torch.randperm` function and observing the impact on predictions. RASI is defined as $RASI = \sum_{(\mathbf{x},y) \in \hat{\mathcal{D}}_{\text{Test}}} \mathbf{1}\{f_{\theta'}(\mathbf{x}) = f_{\theta'}(\mathbf{x}_{\text{Shuffle}})\}$, measuring the proportion of unchanged predictions. When calculating RASI, we shuffle the unlearned feature ten times and take the average of the RASI for the final result.

***SRI&SDI***: Using explainable AI techniques, we assess the significance of the unlearned feature via SHapley Additive exPlanations (SHAP) [22]. Two metrics are defined: the *SHAP retention index* (SRI) and the *SHAP distance-to-zero index* (SDI). SRI quantifies the **relative importance** of the unlearned feature after unlearning, calculated as $SRI = \frac{\kappa_j(f_{\theta'})}{\kappa_j(f_\theta)}$. SDI measures the absolute deviation of the SHAP value from zero, expressed as $SDI = |\kappa_j(f_{\theta'}) - 0|$.

Further, privacy and fairness metrics are considered. For the former, a variant of membership inference attack (MIA) [30] at the feature level is calculated in the following steps. First, we create an attack dataset. For each sample, we query $f_{\theta'}$ and $g_\phi$ to get their outputs that are then labeled as *Unlearned* and *Retrained*, respectively. An attack model, i.e., a Random Forest binary classifier, is further trained on the built attack dataset, with the aim of learning the subtle differences between the outputs from $f_{\theta'}$ and $g_\phi$. The final attack accuracy is the evaluation metric regarding privacy. An accuracy value near 50% indicates that the attack model cannot do better than random guessing, implying that the unlearned model is indistinguishable from the model retrained from scratch. For the fairness metric, we introduce demographic parity (DP) and equalized odds (EO), whose detailed descriptions are provided in Appendix F. For a successful unlearning, the fairness score of the unlearned model should be very close to that of the retrained-from-scratch model.

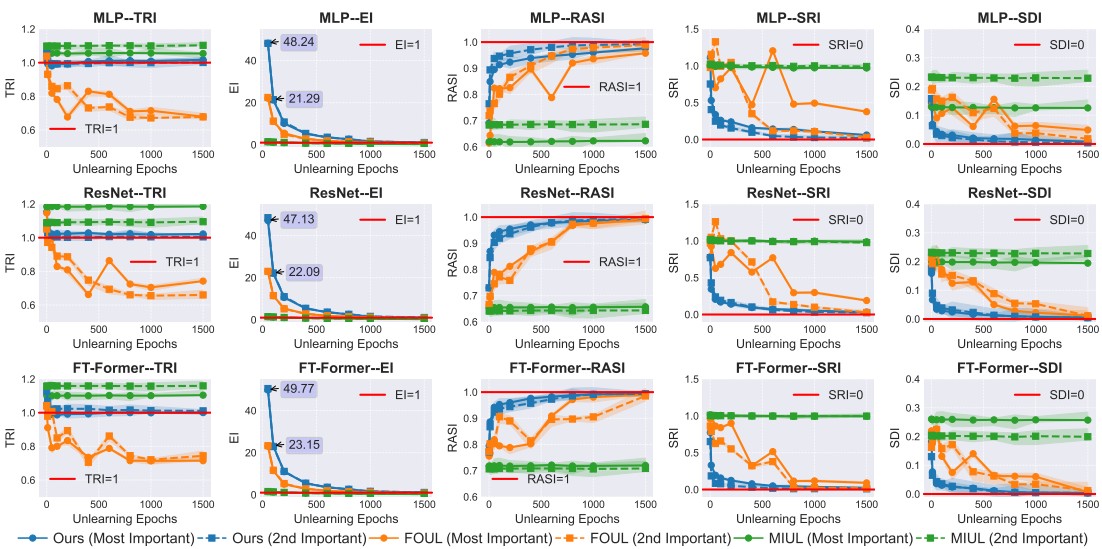

Figure 2: Single-feature unlearning evaluation results of the *Magic* tabular dataset.

## 5.2 Experimental Settings and Results

***Datasets – Tabular***: This study employs six OpenML datasets [11]: *Magic*, *Credit*, and *Cali* for the single-feature unlearning setting (i.e., unlearning one feature), and *Eye*, *Comp*, and *Pol*—which contain more features—to explore multi-feature unlearning. Detailed descriptions of these datasets are provided in Appendix G.

***Dataset – Image***: We further extend our unlearning method to an image classification task using the CelebA [21], a large-scale dataset of over 200K images. The key difference in unlearning between tabular and computer vision (CV) datasets lies in *feature representation*. In tabular data, features are explicitly structured as columns, allowing direct shuffling. In contrast, CV datasets embed features – like nose shape in CelebA – within images, without explicit separation into variables. Hence, we focus on shuffling specific visual features, including nose and eyes, to achieve unlearning for image classification tasks. Examples of processed images with shuffled feature are shown in Appendix H.

***Baselines***: Feature unlearning is an emerging research area with limited prior work. Two baselines, as discussed in Section 2, are included in our experiments: 1) *First-Order Unlearning (FOUL)* [32]: This method applies small permutations to the unlearned feature and leverages the first-order gradient differences between the permuted and original inputs to update the neural network and 2) *Mutual Information Unlearning (MIUL)* [15]: This approach reduces the learned information of the unlearned feature and the label using mutual information estimation while preserving others. It requires training three additional neural networks for mutual information estimation.

***Neural Architectures and Training Parameters***: All experiments are conducted on an NVIDIA V100 GPU. For tabular unlearning, we use a standard MLP and two more complex architectures: ResNet [17] and FT-Transformer [12]. During training, the original model $f_\theta$ is trained for 1500 epochs. Models trained from scratch run for 2000 epochs to ensure convergence (see training loss curves in Appendix J). For our approach and the two baselines, unlearning is performed over 1 to 1500 epochs, with selected values at 1, 10, 50, 100, ..., 1000, and 1500. For the CV task, we leverage the Vision Transformer (ViT) [9] backbone with an MLP classifier head. Model trained from scratch undergoes training for 100 epochs. Unlearning spans 1 to 100 epochs, including specific values at 1, 20, 30, ..., 100. To mitigate randomness, each model is trained *ten times*. Detailed configurations are provided in Appendix I.

### 5.2.1 Results of Single-Feature Unlearning

***Tabular Task– Unlearning Features with Different Feature Importance***: To examine the effectiveness of our method in unlearning different features with varying levels of importance, we perform unlearning on **the most, second most important features**. Each feature's importance is calculated and represented as their Shapley values, with details provided in Appendix K. For brevity, the results

Table 1: Averaged MIA accuracy comparison (optimal: 0.5 ).

| Method \ Epoch | 1 | 10 | 50 | 100 | 200 | 400 | 600 | 800 | 1000 | 1500 |
|---|---|---|---|---|---|---|---|---|---|---|
| Ours | 0.544 | 0.520 | 0.496 | 0.509 | 0.525 | 0.528 | 0.525 | 0.520 | 0.512 | 0.511 |
| FOUL | 0.574 | 0.620 | 0.603 | 0.588 | 0.569 | 0.651 | 0.751 | 0.763 | 0.763 | 0.763 |
| MIUL | 0.651 | 0.596 | 0.597 | 0.597 | 0.605 | 0.603 | 0.595 | 0.612 | 0.594 | 0.614 |

of unlearning the most and the second most important features within the *Magic* dataset are depicted in Fig. 2. The full results of single-feature unlearning of each used tabular dataset, in terms of TRI, EI, RASI, SRI, and SDI, are provided in Appendix L. Additionally, the results of averaged feature-level MIA accuracy across datasets are provided in Table 1. The fairness-related results are provided in Appendix O.

Our results demonstrate that **our approach consistently achieves TRI values close to 1 across all datasets for both various neural architectures and various features**, indicating that the accuracy of the unlearned model nearly matches that of a model trained from scratch. In contrast, the baseline methods exhibit different behaviors: the *FOUL* method experiences accuracy degradation, while the *MIUL* maintains the same accuracy level regardless of the increase in unlearning epochs. Our method empirically proves that it can achieve TRI values around one within a limited number of unlearning epochs.

Additionally, our approach is significantly more efficient than retraining from scratch, as demonstrated by its high EI values. Rather than requiring full retraining until convergence, *our method achieves a TRI value close to one using nearly 200 unlearning epochs across all datasets, making it approximately **10 times faster than training from scratch***. In contrast, both *FOUL* and *MIUL* demand greater computational resources, particularly *MIUL*, which necessitates training additional neural networks for mutual information estimation.

For the RASI metric, our method almost consistently achieves the highest values, indicating that the unlearned feature has been more effectively forgotten compared to the baseline methods. This observation is further reinforced by the SHAP-based evaluations (SRI and SDI), which provide a more detailed assessment of the feature's importance (or dependency with the label) after unlearning. Our approach achieves the lowest SRI values, confirming that the **relative importance of the unlearned feature is significantly diminished**. Additionally, the low SDI values indicate that the absolute contribution of the unlearned feature is reduced to near zero, meaning that even when the feature is present, it no longer influences model predictions. Moreover, we further evaluate our method's effectiveness when unlearning features that are **highly-correlated with others**. The correlation of each two features is calculated via the Pearson correlation coefficient. The resulted correlation heatmaps are provided in Appendix M. We identify one feature pair as highly-correlated features if the Pearson correlation coefficient is greater than 0.8. As shown in Appendix N, the results of unlearning these identified features across all neural architectures and tabular datasets are consistent with the outcomes of unlearning features with different importance, further justifying the effectiveness and robustness of our unlearning method. Furthermore, as shown in Table 1 and Appendix O, the evaluations on MIA, DP, and EO demonstrate the effectiveness of our unlearning methods in terms of privacy and fairness.

*Image Task*: We extend our unlearning method to the CelebA dataset for image classification, targeting the removal of visual features – *nose* and *eyes*. Classification classes include *gender*, *big nose*, *pointy nose*, *eyeglasses*, and *narrow eyes*. The evaluation results across all metrics are detailed in Appendix P, where our method still significantly outperforms baselines and effectively achieves the unlearning goal.We attained an average TRI of 97.31% after 20 epochs and 99.12% after 100 epochs. The strong performance in this image dataset highlights the broader applicability of our method to real-world high dimensional complex scenarios.

### 5.2.2 Results of Multi-Feature Unlearning

We further evaluate the capability of our approach for unlearning multiple features for both tabular and CV datasets. For the tabular datasets, we unlearned more than half of the features in each dataset (with results provided in Appendix Q), and for the CV tasks, we simultaneously unlearned both the *nose* and *eyes* features. Fig. 3 shows evaluation results of unlearning both *Nose* and *Eyes* when

performing classification on label class *Gender*. As shown in Appendix Q and Fig. 3, the request to unlearn a large number of features for tabular datasets, as well as multiple visual features, does not compromise the performance of our approach and consistently outperforms the two baseline methods.

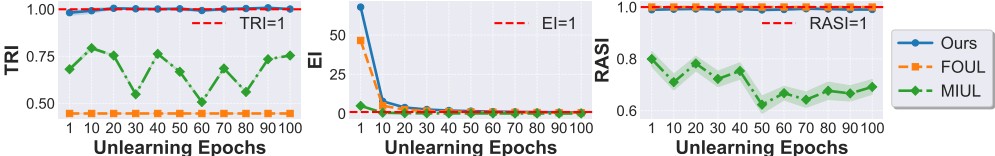

Figure 3: Evaluation results of unlearning *Nose* and *Eyes* for label class *Gender*.

## 6 Limitations and Conclusion

Similar to most machine unlearning techniques in the literature, our approach requires prior knowledge of the data to be removed, specifically access to the training data. Detecting privacy leaks in learning models, which is a complex issue, falls outside the scope of this research. In this paper, we present a straightforward yet effective method for feature unlearning through shuffling on features designated for removal. Despite its simplicity, this approach can theoretically achieve outcomes comparable to retraining a model from scratch under mild assumptions. Extensive empirical evaluations across various datasets demonstrate that our method can simultaneously unlearn multiple features effectively, while maintaining high accuracy and strong generalization capabilities for the remaining features. In summary, our study provides a practical, efficient, and theoretically sound approach to feature unlearning that could significantly impact how machine learning models are updated and maintained, particularly in light of increasing data privacy concerns.

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

# A   Empirical Distribution Convergence Between $X_j^\pi$ and $\mathbf{X}_{-j}$

*Proof.* As $\mathcal{D}$ is stationary and ergodic, the coordinate processes $\{x_{i,j}\}_{i=1}^n$ and $\{\mathbf{x}_{i,-j}\}_{i=1}^n$ both satisfy the *Birkhoff's ergodic theorem* [4], which means that, for each measurable $A \subseteq \mathcal{X}_j$ and $B \subseteq \mathcal{X}_{-j}$:

$$\hat{P}_{X_j}^{(n)}(A) = \frac{1}{n} \sum_{i=1}^n \mathbf{1}\{x_{i,j} \in A\} \xrightarrow[n \to \infty]{\text{a.s.}} P_{X_j}(A), \tag{21}$$

$$\hat{P}_{\mathbf{X}_{-j}}^{(n)}(B) = \frac{1}{n} \sum_{i=1}^n \mathbf{1}\{\mathbf{x}_{i,-j} \in B\} \xrightarrow[n \to \infty]{\text{a.s.}} P_{\mathbf{X}_{-j}}(B). \tag{22}$$

Hence, $\hat{P}_{X_j}^{(n)} \to P_{X_j}$ and $\hat{P}_{\mathbf{X}_{-j}}^{(n)} \to P_{\mathbf{X}_{-j}}$ almost surely.

The shuffled empirical distribution of a rectangle $A \times B$ is written as

$$\hat{P}_{X_j^\pi, \mathbf{X}_{-j}}^{(n)}(A \times B) = \frac{1}{n} \sum_{i=1}^n \mathbf{1}\{x_{\pi(i),j} \in A, \mathbf{x}_{i,-j} \in B\}. \tag{23}$$

For this empirical distribution, when conditioning on the process $\{x_{i,j}, \mathbf{x}_{i,-j}\}_{i=1}^n$, for each fixed $i$, the index $\pi(i)$ is uniform in $\{1, \dots, n\}$. Therefore, we have

$$\mathbb{E}_\pi[\mathbf{1}\{x_{\pi(i),j} \in A\} | \{x_{i,j}, \mathbf{x}_{i,-j}\}_{i=1}^n] = \hat{P}_{x_j}^{(n)}(A). \tag{24}$$

Multiplying by $\mathbf{1}\{\mathbf{x}_{i,-j} \in B\}$ and summing over $i$ then shows

$$\mathbb{E}_\pi[\hat{P}_{X_j^\pi, \mathbf{X}_{-j}}^{(n)}(A \times B) | \{x_{i,j}, \mathbf{x}_{i,-j}\}_{i=1}^n] = \hat{P}_{X_j}^{(n)}(A) \hat{P}_{\mathbf{X}_{-j}}^{(n)}(B). \tag{25}$$

We then un-condition the above expectation using the *law of total expectation*, written as

$$\begin{aligned}
\mathbb{E}_\pi[\hat{P}_{X_j^\pi, \mathbf{X}_{-j}}^{(n)}(A \times B)] &= \mathbb{E}_{\{x_{i,j}, \mathbf{x}_{i,-j}\}_{i=1}^n}[\hat{P}_{X_j}^{(n)}(A) \hat{P}_{\mathbf{X}_{-j}}^{(n)}(B)] \\
&= \hat{P}_{X_j}^{(n)}(A) \hat{P}_{\mathbf{X}_{-j}}^{(n)}(B).
\end{aligned} \tag{26}$$

Notably, the last term is derived based on the fact that the product of two marginal probabilities is a constant for a fixing $(x_{\pi(i),j}, \mathbf{x}_{i,-j})$, i.e., no more randomness in $\pi$ for the marginals.

Given Eq. (26), by defining the difference between the shuffled empirical distributions and the product of two marginal empirical distributions, i.e.,

$$\Omega_n(A, B) := \hat{P}_{X_j^\pi, \mathbf{X}_{-j}}^{(n)}(A \times B) - \hat{P}_{X_j}^{(n)}(A) \hat{P}_{\mathbf{X}_{-j}}^{(n)}(B). \tag{27}$$

We see that $\mathbb{E}_\pi[\Omega_n(A, B)] = 0$, i.e., $\Omega_n(A, B)$ is mean zero. Next, we aim to show that $\Omega_n(A, B)$ is *small with high probability*.

To prove that, we first define

$$S_n(A, B) = \sum_{i=1}^n \mathbf{1}\{x_{\pi(i),j} \in A, \mathbf{x}_{i,-j} \in B\}. \tag{28}$$

Then, the shuffled empirical distribution $\hat{P}_{X_j^\pi, \mathbf{X}_{-j}}^{(n)}(A \times B)$ can be expressed as $\frac{1}{n} S_n(A, B)$. Changing $\pi$ in one position affects at most *two* indicators in $S_n$. Hence, the rewritten shuffled empirical distribution $\frac{1}{n} S_n(A, B)$ satisfies *McDiarmid's inequality* [25] – for some $c, \epsilon > 0$,

$$P(|\frac{1}{n} S_n(A, B) - \mathbb{E}_\pi[\frac{1}{n} S_n(A, B)]| > \epsilon) \le 2 \exp\left(-\frac{2\varepsilon^2}{\sum_{i=1}^n c_i^2}\right). \tag{29}$$

Given Eq. (26), we have $\mathbb{E}_\pi[\frac{1}{n} S_n(A, B)] = \hat{P}_{X_j}^{(n)}(A) \hat{P}_{\mathbf{X}_{-j}}^{(n)}(B)$. The above McDiarmid's inequality can be rewritten as

$$\begin{aligned}
&P(|\hat{P}_{X_j^\pi, \mathbf{X}_{-j}}^{(n)}(A \times B) - \hat{P}_{X_j}^{(n)}(A) \hat{P}_{\mathbf{X}_{-j}}^{(n)}(B)| > \epsilon) \\
&= P(|\Omega_n(A, B)| > \epsilon) \le 2 \exp\left(-\frac{2\varepsilon^2}{\sum_{i=1}^n c_i^2}\right).
\end{aligned} \tag{30}$$

Then, summing over $n$, we have

$$\sum_{n=1}^{\infty} P(|\Omega_n(A, B)| > \epsilon) \leq \infty, \tag{31}$$

which satisfies the *Borel-Cantelli* lemma [36, 7], resulting in $\Omega_n(A, B) \to 0$ almost surely for each fixed $A \times B$. By combining it with the *Birkhoff's ergodic theorem* in Eq. (21) and (22) (i.e., the empirical marginal distribution almost surely converges to its marginal distribution), we have

$$\hat{P}^{(n)}_{X_j^\pi, \mathbf{X}_{-j}}(A \times B) = \hat{P}^{(n)}_{X_j}(A) \hat{P}^{(n)}_{\mathbf{X}_{-j}}(B) + \Omega_n(A, B)$$
$$\xrightarrow[n \to \infty]{\text{a.s.}} P_{X_j}(A) \times P_{\mathbf{X}_{-j}}(B). \tag{32}$$

So far, we have *point-wise* almost surely convergence for each rectangle $A \times B$. To further show the shuffled empirical distribution $\hat{P}^{(n)}_{X_j^\pi, \mathbf{X}_{-j}}(A \times B)$ almost surely converges to $P_{X_j} \times P_{\mathbf{X}_{-j}}$ on *all* measurable sets, we choose a countable family of rectangles $\{A_l \times B_l\}$ that generates the product $\sigma$-algebra on $\mathcal{X}_j \times \mathcal{X}_{-j}$. By the same exponential tail bound in Eq. (30), a union-of-Borel-Cantelli shows $|\Omega_n(A_l, B_l)| \to 0$ simultaneously for all $l$ with the probability of 1.

As these rectangles form (or generate) a $\pi$-system for $\mathcal{X}_j \times \mathcal{X}_{-j}$, the standard measure-theoretic uniqueness results implies

$$\hat{P}^{(n)}_{X_j^\pi, \mathbf{X}_{-j}} \xrightarrow[n \to \infty]{\text{a.s.}} P_{X_j} \times P_{\mathbf{X}_{-j}}. \tag{33}$$

$\square$

# B  Proof Independence between $X_j^\pi$ and $(Y, \mathbf{X}_{-j})$

*Proof.* In discrete case, the relationship between the empirical joint probability mass function (PMF) and the empirical joint probability measure can be written as $\forall (x_j^\pi, y) \in \mathcal{X}_j \times \mathcal{Y}$:

$$p_{X_j^\pi,Y}^{(n)}(x_j^\pi, y) = \hat{P}_{X_j^\pi,Y}^{(n)}(X_j^\pi = x_j^\pi, Y = y). \tag{34}$$

Similarly, the relationship between the product of empirical marginal PMFs and the product of empirical marginal probability measures can be written as

$$p_{X_j^\pi}(x_j^\pi) p_Y(y) = P_{X_j^\pi}(X_j^\pi = x_j^\pi) P_Y(Y = y). \tag{35}$$

Given $\hat{P}_{X_j^\pi,Y}^{(n)} \xrightarrow[n\to\infty]{\text{a.s.}} P_{X_j} \times P_Y$, we have, $\forall (x_j^\pi, y) \in \mathcal{X}_j \times \mathcal{Y}$

$$\hat{P}_{X_j^\pi,Y}^{(n)}(X_j^\pi = x_j^\pi, Y = y) \xrightarrow[n\to\infty]{P} {}_{X_j^\pi}(X_j^\pi = x_j^\pi) P_Y(Y = y). \tag{36}$$

Hence, the PMF $p_{X_j^\pi,Y}^{(n)}(x_j^\pi, y)$ almost surely converges pointwise to $p_{X_j^\pi}(x_j^\pi) p_Y(y)$ for $\forall (x_j^\pi, y) \in \mathcal{X}_j \times \mathcal{Y}$, expressed as

$$p_{X_j^\pi,Y}^{(n)}(x_j^\pi, y) \xrightarrow[n\to\infty]{\text{a.s.}} p_{X_j^\pi}(x_j^\pi) p_Y(y). \tag{37}$$

Similarly, we can have

$$\hat{p}_{X_j^\pi}^{(n)}(x_j^\pi) \xrightarrow[n\to\infty]{\text{a.s.}} p_{X_j^\pi}(x_j^\pi), \quad \hat{p}_Y^{(n)}(y) \xrightarrow[n\to\infty]{\text{a.s.}} p_Y(y). \tag{38}$$

In continuous case, we assume that both empirical and actual probabilities measures, including $\hat{P}_{X_j^\pi,Y}^{(n)}, \hat{P}_{X_j^\pi}^{(n)}, \hat{P}_Y^{(n)}, P_{X_j^\pi,Y}, P_{X_j^\pi}, P_Y$, are absolutely continuous w.r.t. Lebesgue measure $\lambda$. Hence, there exists PDFs $\hat{p}_{X_j^\pi,Y}^{(n)}(x_j^\pi, y), \hat{p}_{X_j^\pi}^{(n)}(x_j^\pi), \hat{p}_Y^{(n)}(y), p_{X_j^\pi,Y}(x_j^\pi, y), p_{X_j^\pi}(x_j^\pi), p_Y(y)$ such that

$$\hat{P}_{X_j^\pi,Y}^{(n)} = \int_A \hat{p}_{X_j^\pi,Y}^{(n)}(x_j^\pi, y) d\lambda(x_j^\pi, y), \quad \forall \text{Borel Set } A \subseteq \mathcal{X}_j \times \mathcal{Y}, \tag{39}$$

$$\hat{P}_{X_j^\pi}^{(n)} = \int_A \hat{p}_{X_j^\pi}^{(n)}(x_j^\pi) d\lambda(x_j^\pi), \quad \forall \text{Borel Set } A \subseteq \mathcal{X}_j, \tag{40}$$

$$\hat{P}_Y^{(n)} = \int_A \hat{p}_Y^{(n)}(y) d\lambda(y), \quad \forall \text{Borel Set } A \subseteq \mathcal{Y}, \tag{41}$$

$$P_{X_j^\pi,Y} = \int_A p_{X_j^\pi,Y}(x_j^\pi, y) d\lambda(x_j^\pi, y), \quad \forall \text{Borel Set } A \subseteq \mathcal{X}_j \times \mathcal{Y}, \tag{42}$$

$$P_{X_j^\pi} = \int_A p_{X_j^\pi}(x_j^\pi) d\lambda(x_j^\pi), \quad \forall \text{Borel Set } A \subseteq \mathcal{X}_j, \tag{43}$$

$$P_Y = \int_A p_Y(y) d\lambda(y), \quad \forall \text{Borel Set } A \subseteq \mathcal{Y}. \tag{44}$$

As we already have $\hat{P}_{X_j^\pi,Y}^{(n)} \xrightarrow[n\to\infty]{\text{a.s.}} P_{X_j} \times P_Y$, therefore, it ensures the total variation convergence expressed as

$$\|\hat{P}_{X_j^\pi,Y}^{(n)} - P_{X_j} \times P_Y\|_{\text{TV}} = \frac{1}{2} \int |\hat{p}_{X_j^\pi,Y}^{(n)} - p_{X_j^\pi} p_Y| d\lambda \xrightarrow[n\to\infty]{\text{a.s.}} 0. \tag{45}$$

That is to say $\|\hat{p}_{X_j^\pi,Y}^{(n)} - p_{X_j} p_Y\|_1 \xrightarrow[n\to\infty]{\text{a.s.}} 0$ in $L^1(\lambda)$. Similarly, we can have $\|\hat{p}_{X_j^\pi}^{(n)} - p_{X_j}\|_1 \xrightarrow[n\to\infty]{\text{a.s.}} 0$ in $L^1(\lambda)$ and $\|\hat{p}_Y^{(n)} - p_Y\|_1 \xrightarrow[n\to\infty]{\text{a.s.}} 0$ in $L^1(\lambda)$. Based on the Scheffé's lemma [28], from $L^1$ convergence of a sequence of non-negative functions to a non-negative limit, it follows that

$$\hat{p}_{X_j^\pi,Y}^{(n)} \xrightarrow[n\to\infty]{\text{a.s.}} p_{X_j} p_Y, \tag{46}$$

$$\hat{p}_{X_j^\pi}^{(n)} \xrightarrow[n\to\infty]{\text{a.s.}} p_{X_j}, \tag{47}$$

$$\hat{p}_Y^{(n)} \xrightarrow[n\to\infty]{\text{a.s.}} p_Y. \tag{48}$$

Therefore, we now have

$$\hat{p}^{(n)}_{X^\pi_j, Y}(x^\pi_j, y) \xrightarrow[n\to\infty]{\text{a.s.}} p_{X_j}(x_j) p_Y(y) = p_{X^\pi_j}(x^\pi_j) p_Y(y). \tag{49}$$

Analogously, we also have

$$\hat{p}^{(n)}_{X^\pi_j, \mathbf{X}_{-j}}(x^\pi_j, \mathbf{x}_{-j}) \xrightarrow[n\to\infty]{\text{a.s.}} p_{X_j}(x_j) p_{\mathbf{X}_{-j}}(\mathbf{x}_{-j}) = p_{X^\pi_j}(x^\pi_j) p_{\mathbf{X}_{-j}}(\mathbf{x}_{-j}). \tag{50}$$

For every measurable sets $A \subseteq \mathcal{X}^\pi_j$ and $B \subseteq \mathcal{Y}$, we have

$$p(X^\pi_j \in A, Y \in B) = \int_B \int_A p_{X^\pi_j, Y}(x^\pi_j, y) dx^\pi_j dy = \int_B p_Y(y) dy \int_A p_{X^\pi_j}(x^\pi_j) d(x^\pi_j). \tag{51}$$

Hence, we have

$$p(X^\pi_j \in A, Y \in B) = p(X^\pi_j \in A) p(Y \in B), \tag{52}$$

and therefore $X^\pi_j \perp\!\!\!\perp Y$. Similarly, we can have $X^\pi_j \perp\!\!\!\perp \mathbf{X}_{-j}$. Finally, when $n \to \infty$, we almost surely have

$$X^\pi_j \perp\!\!\!\perp (Y, \mathbf{X}_{-j}). \tag{53}$$

$$\square$$

## C   Proof of $(\epsilon - \delta)$-Close of $f_{\theta'}$ and $g_\phi$ Under Non-Strongly Convex Loss Function $\ell$

*Proof.* If $\ell$ is not strongly convex but bounded, the ERM problem may have multiple global minimizers. Therefore, for each $f_{\theta'}$ derived from Eq. (18), there exists a $g_\phi$ that their achieved empirical risks are $(\epsilon - \delta)$-close.

We here propose a mild assumption that

$$|f_{\theta'} - g_\phi| \leq \omega |\ell(f_{\theta'}, y) - \ell(g_\phi, y)|, \tag{54}$$

where $\omega$ represents a scaling factor that bounds how sensitive the model's output or parameters are to changes in loss function.

By taking expectations on both sides of Eq. (54), we have

$$\mathbb{E}\left[|f_{\theta'} - g_\phi|\right] \leq \omega \mathbb{E}\left[|\ell(f_{\theta'}, y) - \ell(g_\phi, y)|\right]. \tag{55}$$

Based on Eq. (17), the RHS of the above inequality, i.e., empirical loss difference, is bounded by $6\epsilon_n + 2\alpha$. Therefore, we have

$$\mathbb{E}\left[|f_{\theta'} - g_\phi|\right] \leq \omega\left(6\epsilon_n + 2\alpha\right). \tag{56}$$

To further bound the difference of two models, i.e., $|f_{\theta'} - g_\phi|$, we apply the *Markov's inequality*:

$$P\left(|f_{\theta'} - g_\phi| > \epsilon\right) \leq \frac{\mathbb{E}\left[|f_{\theta'} - g_\phi|\right]}{\epsilon} \leq \frac{\omega(6\epsilon_n + 2\alpha)}{\epsilon}. \tag{57}$$

To ensure such probability is at most $\delta$, we set

$$\frac{\omega(6\epsilon_n + 2\alpha)}{\epsilon} \leq \delta, \tag{58}$$

which is rearranged as

$$\epsilon \geq \frac{\omega(6\epsilon_n + 2\alpha)}{\delta}. \tag{59}$$

Therefore, by choosing $\epsilon = \frac{\omega(6\epsilon_n + 2\alpha)}{\delta}$, we have $f_{\theta'}$ and $g_\phi$ are $(\epsilon - \delta)$-close.

□

# D  Empirical Mutual Information Convergence

*Proof.* The empirical mutual information between $X_j^\pi$ and $Y$ can be defined as

$$I_n(X_j^\pi; Y) := \mathbb{E}_{(x_j^\pi, y) \sim \hat{P}_{X_j^\pi, Y}^{(n)}} \left[ \log \frac{\hat{p}_{X_j^\pi, Y}^{(n)}(x_j^\pi, y)}{\hat{p}_{X_j^\pi}^{(n)}(x_j^\pi) \hat{p}_Y^{(n)}(y)} \right]. \tag{60}$$

Given the independence between $X_j^\pi$ and $(Y, \mathbf{X}_{-j})$ proved in Corollary 4.2, we have

$$\hat{p}_{X_j^\pi, Y}^{(n)}(x_j^\pi, y) \xrightarrow[n \to \infty]{\text{a.s.}} p_{X_j}(x_j) p_Y(y) = p_{X_j^\pi}(x_j^\pi) p_Y(y), \tag{61}$$

$$\hat{p}_{X_j^\pi}^{(n)}(x_j^\pi) \xrightarrow[n \to \infty]{\text{a.s.}} \hat{p}_{X_j^\pi}^{(n)}, \tag{62}$$

$$\hat{p}_Y^{(n)}(y) \xrightarrow[n \to \infty]{\text{a.s.}} p_Y(y). \tag{63}$$

Therefore, the ratio inside the logarithm operator of mutual information almost surely converges to 1, i.e.,

$$\frac{\hat{p}_{X_j^\pi, Y}^{(n)}(x_j^\pi, y)}{\hat{p}_{X_j^\pi}^{(n)}(x_j^\pi) \hat{p}_Y^{(n)}(y)} \xrightarrow[n \to \infty]{\text{a.s.}} 1. \tag{64}$$

For large $n$, this ratio stays in $[\delta, \frac{1}{\delta}]$ for some $\delta > 0$. Therefore, we have

$$\left| \log \frac{\hat{p}_{X_j^\pi, Y}^{(n)}(x_j^\pi, y)}{\hat{p}_{X_j^\pi}^{(n)}(x_j^\pi) \hat{p}_Y^{(n)}(y)} \right| \leq \max \left\{ |\log \delta|, \log \left( \frac{1}{\delta} \right) \right\}, \tag{65}$$

indicating that $\log \frac{\hat{p}_{X_j^\pi, Y}^{(n)}(x_j^\pi, y)}{\hat{p}_{X_j^\pi}^{(n)}(x_j^\pi) \hat{p}_Y^{(n)}(y)}$ is *uniformly bounded* for large $n$. By leveraging the dominated convergence theorem [2], we have

$$\mathbb{E}_{(x_j^\pi, y) \sim \hat{P}_{X_j^\pi, Y}^{(n)}} \left[ \log \frac{\hat{p}_{X_j^\pi, Y}^{(n)}(x_j^\pi, y)}{\hat{p}_{X_j^\pi}^{(n)}(x_j^\pi) \hat{p}_Y^{(n)}(y)} \right] = I_n(X_j^\pi; Y) \xrightarrow[n \to \infty]{\text{a.s.}} 0. \tag{66}$$

Analogously, we can have

$$I_n(X_j^\pi, \mathbf{X}_{-j}) \xrightarrow[n \to \infty]{\text{a.s.}} 0, \tag{67}$$

$$I_n(X_j^\pi, \theta') \xrightarrow[n \to \infty]{\text{a.s.}} 0. \tag{68}$$

$\square$

# E  Almost Surely Equal Shapley Value between Unlearned Model and Retrained Model

*Proof.* We treat each feature as a "player" in a cooperative game. For a subset $S \subseteq \{1, \ldots, m\}$ of features, we define $v(S)$ as the performance (e.g., accuracy) of model trained with *only* features in $S$.

**Definition E.1** (*Shapley Value of the $i$-th Feature*)**.** The *Shapley value* of the $i$-th feature ($i \in \{1, \ldots, m\}$) is

$$\kappa_i = \sum_{S \subseteq \{1, \ldots, m\} \setminus \{i\}} \frac{|S|!\, (m - |S| - 1)!}{m!} \left[ v(S \cup \{i\}) - v(S) \right],$$

where $\phi_i$ is the *expected marginal contribution* of the $i$-th feature to the performance, averaging over all subsets $S$ that do not contain $i$.

Given that $X_j^{\pi}$ is independent with $Y$, we have, for any subset of features $S \subseteq \{1, \ldots, m\}$:

$$v_{f_{\theta'}}(S \cup \{j\}) = v_{f_{\theta'}}(S). \tag{69}$$

For the retrained model, we define its corresponding performance as $v_{g_\phi}(S)$. If a subset $S$ does not contain $j$, we train only on those features in $S$. If $S$ does contain $j$, then effectively $j$ is not available. Therefore, $S \cup \{j\}$ is the same as $S$ in terms of actual features used for model $g_\phi$, which means

$$v_{g_\phi}(S \cup \{j\}) = v_{g_\phi}(S). \tag{70}$$

Given Eq. (69) and (70), $\phi_j$ equals zero in both models, showing they coincide for the "unlearned" feature (i.e., the $j$-th feature).

This is the crux: we want to see that each unchanged (or shuffled) feature retains the *same* Shapley value in both models.

Consider any subset $S \subseteq \{1, \ldots, m\} \setminus \{i\}$ which *does not* contain $i$. The Shapley value for the $i$-th feature derived from both model can be simplified as $v_{f_{\theta'}}(S \cup \{i\}) - v_{f_{\theta'}}(S)$ and $v_{g_\phi}(S \cup \{i\}) - v_{g_\phi}(S)$, respectively. We need to show that these two differences are the same for all $S$, which can be divided into the following two cases:

*Case 1*: If $j \notin S$ and $j \notin (S \cup \{i\})$, then $f_{\theta'}$ is using exactly the same features as $g_\phi$ (because $j$ is absent in both). Thus,

$$v_{f_{\theta'}}(S \cup \{i\}) = v_{g_\phi}(S \cup \{i\}), \tag{71}$$

$$v_{f_{\theta'}}(S) = v_{g_\phi}(S), \tag{72}$$

resulting in the same Shapley value.

*Case 2*: If $j \in S$ (or $j \in S \cup \{i\}$), recall that adding the *worthless* $j$-th feature does not affect the performance of $f_{\theta'}$. Similarly, in the $g_\phi$, the $j$-th feature is forcibly absent. Thus effectively $S$ is *the same* whether or not it nominally includes $j$, indicating:

$$S' = S \setminus \{j\}, \tag{73}$$

$$(S \cup \{i\})' = (S \cup \{i\}) \setminus \{j\}. \tag{74}$$

Hence, we can identify

$$v_{f_{\theta'}}\left(S \cup \{i\}\right) = v_{f_{\theta'}}\left(S' \cup \{i\}\right), \tag{75}$$

$$v_{g_\phi}\left(S \cup \{i\}\right) = v_{g_\phi}\left(S' \cup \{i\}\right), \tag{76}$$

which is also likewise for $v_{f_{\theta'}}(S)$ v.s. $v_{g_\phi}(S')$, as the $j$-th is worthless in $f_{\theta'}$.

In *all* cases, for any subset $S$ *not* containing $i$, we have

$$v_{f_{\theta'}}(S \cup \{i\}) - v_{f_{\theta'}}(S) = v_{g_\phi}(S \cup \{i\}) - v_{g_\phi}(S). \tag{77}$$

Therefore, each marginal contribution term in the Shapley-value sum for the $i$-th feature is identical between $f_{\theta'}$ and $g_\phi$. Summing over all subsets $S$ (with appropriate combinatorial weights), we almost surely conclude that $\forall i \in \{1, \ldots, j - 1, j + 1, \ldots, m\}$:

$$\kappa_i(f_{\theta'}) = \kappa_i(g_\phi). \tag{78}$$

$\square$

## F  Descriptions of Fairness Evaluation Metrics

The demographic parity (DP) score is calculated in the following steps. Firstly, we identify several key components related to DP, including

- Unprivileged/Privileged Groups: The group that has historically faced a disadvantage/advantage (e.g., for features like "Race" or "Gender").
- Favorable Outcome: The desired model prediction (e.g., "loan approved").
- Model Predictions: The output labels generated by our unlearned model or the retrained-from-scratch model.

Next, for both the privileged and unprivileged groups, we calculate the rate at which they receive the favorable outcome from models. The formula for the rate is calculated as

$$\text{Rate} = \frac{\text{Number of Individuals in the group who received the favorable outcome}}{\text{Total number of individuals in the group}}. \tag{79}$$

The above rates are denoted as $\text{DP}(\text{Label=Favorable Outcome} \mid \text{Privileged Group})$ and $\text{DP}(\text{Label=Favorable Outcome} \mid \text{Unprivileged Group})$, respectively.

Finally, we calculate the DP differences (as the DP metric used in evaluation) between unprivileged and privileged groups, formulated as

$$\begin{aligned} \text{DP Difference} = {} & \text{DP}(\text{Label=Favorable Outcome} \mid \text{Unprivileged Group}) \\ & - \text{DP}(\text{Label=Favorable Outcome} \mid \text{Privileged Group}). \end{aligned} \tag{80}$$

A DP difference of zero indicates perfectly fair unlearning.

For another fairness metric, i.e., the equalized odds (EO) difference, we calculate the differences of True Positive Rate (TPR), False Positive Rate (FPR) between the privileged and the unprivileged groups. Both metrics, i.e., TPR and FPR, should be as close as to zero for a successful unlearning from the perspective of fairness.

## G Descriptions of Tabular Datasets

A summary of used dataset from OpenML [11, 6] in this paper is presented in Table 2.

Table 2: Summary of used datasets for single-feature unlearning.

| Dataset Name | Number of Samples | Number of Features | Number of Numerical Features | Number of Categorical Features |
|---|---|---|---|---|
| *Magic* | 13376 | 6 | 6 | 0 |
| *Credit* | 16714 | 10 | 10 | 0 |
| *Cali* | 20634 | 8 | 8 | 0 |

- *Magic*: The data are MC generated (see below) to simulate registration of high energy gamma particles in a ground-based atmospheric Cherenkov gamma telescope using the imaging technique. Cherenkov gamma telescope observes high energy gamma rays, taking advantage of the radiation emitted by charged particles produced inside the electromagnetic showers initiated by the gammas, and developing in the atmosphere. This Cherenkov radiation (of visible to UV wavelengths) leaks through the atmosphere and gets recorded in the detector, allowing reconstruction of the shower parameters. The available information consists of pulses left by the incoming Cherenkov photons on the photomultiplier tubes, arranged in a plane, the camera. Depending on the energy of the primary gamma, a total of few hundreds to some 10000 Cherenkov photons get collected, in patterns (called the shower image), allowing to discriminate statistically those caused by primary gammas (signal) from the images of hadronic showers initiated by cosmic rays in the upper atmosphere (background). The link to this dataset is `https://www.openml.org/d/44125`.

- *Credit*: The link to this dataset is `https://www.openml.org/d/44089`.

- *Cali*: This dataset, also known as the California Housing Prices dataset, contains information about housing values in California, collected from the 1990 U.S. Census. It contains features such as including median income, total rooms, total bedrooms, population, and geographical attributes such as latitude and longitude. This link to this dataset is `https://www.openml.org/d/44090`.

A summary of used dataset for multi-feature unlearning from OpenML [11, 6] is shown in Table 3.

- *Eye*: This dataset was published in the Inferring Relevance from Eye Movements Challenge 2005 [18]. The link to the dataset is `https://www.openml.org/d/44157`.

Table 3: Summary of used datasets for multi-feature unlearning.

| Dataset Name | Number of Samples | Number of Features | Number of Numerical Features | Number of Categorical Features | Number of Unlearned Features |
|---|---|---|---|---|---|
| *Eye* | 7608 | 23 | 20 | 3 | 15 |
| *Comp* | 16644 | 17 | 8 | 9 | 8 |
| *Pol* | 10082 | 26 | 26 | 0 | 20 |

- *Comp*: The link to the dataset is `https://www.openml.org/d/44162`.

- *Pol*: The link to the dataset is `https://www.openml.org/d/44122`.

# H    Examples of Processed Images with Shuffled Visual Features

The following example illustrates how an original image is processed for our unlearning method and retrain-from-scratch approach:

- Shuffled nose feature for our unlearning method: The nose region is shuffled while preserving the overall structure of the image.
- Masked nose feature for retraining from scratch: The nose region is masked to remove its influence on the model.

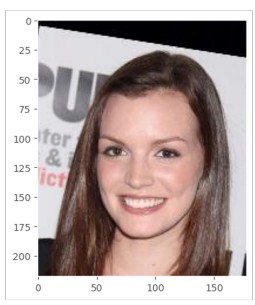 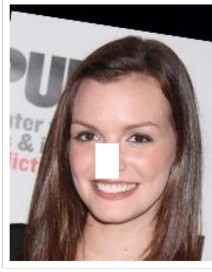 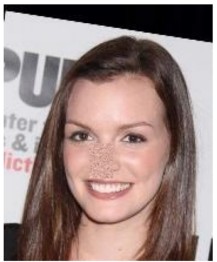

Original Image          Processed Image for          Processed Image with
                        Retrain-from-Scratch         Shuffled Feature

Figure 4: Example of images with shuffled visual features.

# I Configurations of Neural Models and Training Hyperparameter

The batch size is set as $64$ and the learning rate is set as $0.001$. The optimizer used for neural network update is the Adam optimizer and the loss function is the cross-entropy loss function.

For the *FOUL* baseline [32], the scale of permutation on the unlearned feature is set as $0.01$.

For the *MIUL* baseline [34], the training epochs to estimate mutual information is set as $500$. The weight coefficients balancing different mutual information estimations, denoted as $\lambda_1$, $\lambda_2$, and $\lambda_3$, are set as $5$, $5$, and $1$, respectively.

For our unlearning method within the tabular datasets, the MLP model has two hidden layers with the number of neurons of $128$ and $64$. The ResNet model has two residual blocks and a final fully-connected layer for classification outputs. The FT-Transformer model has two transformer blocks with a two-layer MLP head (whose hidden layer dimension is $64$) for classification.

For our unlearning method within the CV dataset, the ViT backbone is imported through the `timm` package – `timm.create_model("vit_base_patch16_224")`. The MLP classifier head has two hidden layers with the number of neurons of $512$ and $128$.

When implementing the SHAP method to calculate the feature importance of unlearned feature, we use the `shap` package developed by [22] and set the background data size and test data size as $1000$ and $100$, respectively.

All codes are implemented in Python 3.10 and PyTorch 1.12.

## J Training Loss Curves of Models Trained From Scratch Under the Single- and Multi-Feature Unlearning Setting for Tabular Datasets

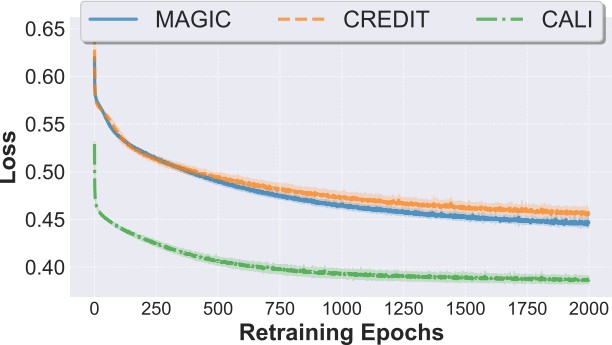

Figure 5: Loss curves of MLP-based models trained from scratch under the single-feature unlearning setting.

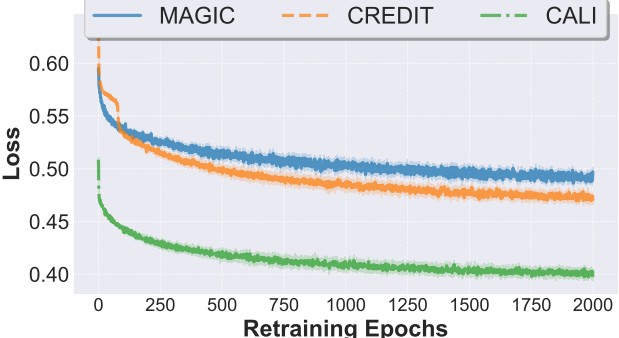

Figure 6: Loss curves of ResNet-based models trained from scratch under the single-feature unlearning setting.

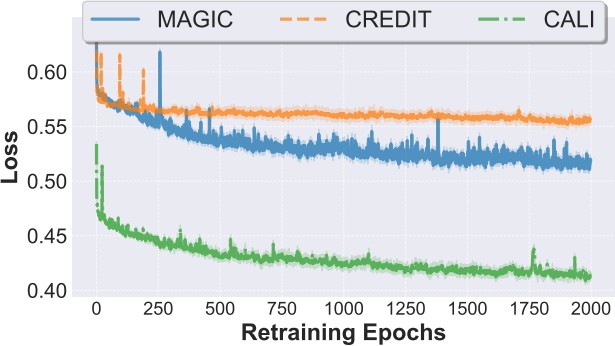

Figure 7: Loss curves of FT-Transformer-based models trained from scratch under the single-feature unlearning setting.

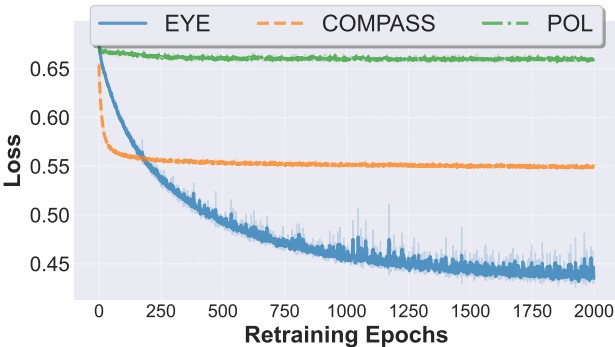

Figure 8: Loss curves of MLP-based models trained from scratch under the multi-feature unlearning setting.

## K  Shapley Values of Each Feature for all Used Tabular Datasets

When implementing the SHAP method to calculate the feature importance of unlearned feature, we use the `shap` package developed by [22] and set the background data size and test data size as 1000 and 100, respectively.

Table 4 shows the Shapley values of each feature in the *Credit* dataset. The most, second most, second last, and last important features are `NumberOfTime30-59DaysPastDueNotWorse`, `NumberOfTimes90DaysLate`, `NumberOfDependents`, `DebtRatio`.

Table 4: Shapley values of each feature in the *Credit* dataset.

| Feature | Shapley Value |
|---|---|
| NumberOfTime30-59DaysPastDueNotWorse | 0.130686 |
| NumberOfTimes90DaysLate | 0.095737 |
| NumberOfTime60-89DaysPastDueNotWorse | 0.083108 |
| NumberOfOpenCreditLinesAndLoans | 0.080779 |
| age | 0.074865 |
| RevolvingUtilizationOfUnsecuredLines | 0.074132 |
| NumberRealEstateLoansOrLines | 0.062857 |
| MonthlyIncome | 0.058103 |
| NumberOfDependents | 0.035402 |
| DebtRatio | 0.001361 |

Table 5 shows the Shapley values of each feature in the *Cali* dataset. The most, second most, second last, and last important features are `Latitude`, `Longitude`, `AveBedrms`, `Population`.

Table 5: Shapley values of each feature in the *Cali* dataset.

| Feature | Shapley Value |
|---|---|
| Latitude | 0.281184 |
| Longitude | 0.248730 |
| MedInc | 0.148034 |
| AveOccup | 0.081746 |
| AveRooms | 0.057758 |
| HouseAge | 0.041526 |
| AveBedrms | 0.021693 |
| Population | 0.019374 |

Table 6 shows the Shapley values of each feature in the *Magic* dataset. The most, second most, second last, and last important features are `fWidth`, `fSize`, `fLength`, `fAsym`.

Table 6: Shapley values of each feature in the *Magic* dataset.

| Feature | Shapley Value |
|---|---|
| fWidth | 0.129222 |
| fSize | 0.101183 |
| fConc | 0.095207 |
| fConc1 | 0.089712 |
| fLength | 0.079942 |
| fAsym | 0.053422 |

## L Full Results of Unlearning Features with Different Feature Importance Under Single-Feature Unlearning Settings

Fig. 9 illustrates the TRI results of unlearning the most, second most, second last, and last important features across the MLP, ResNet, and FT-Transformer models.

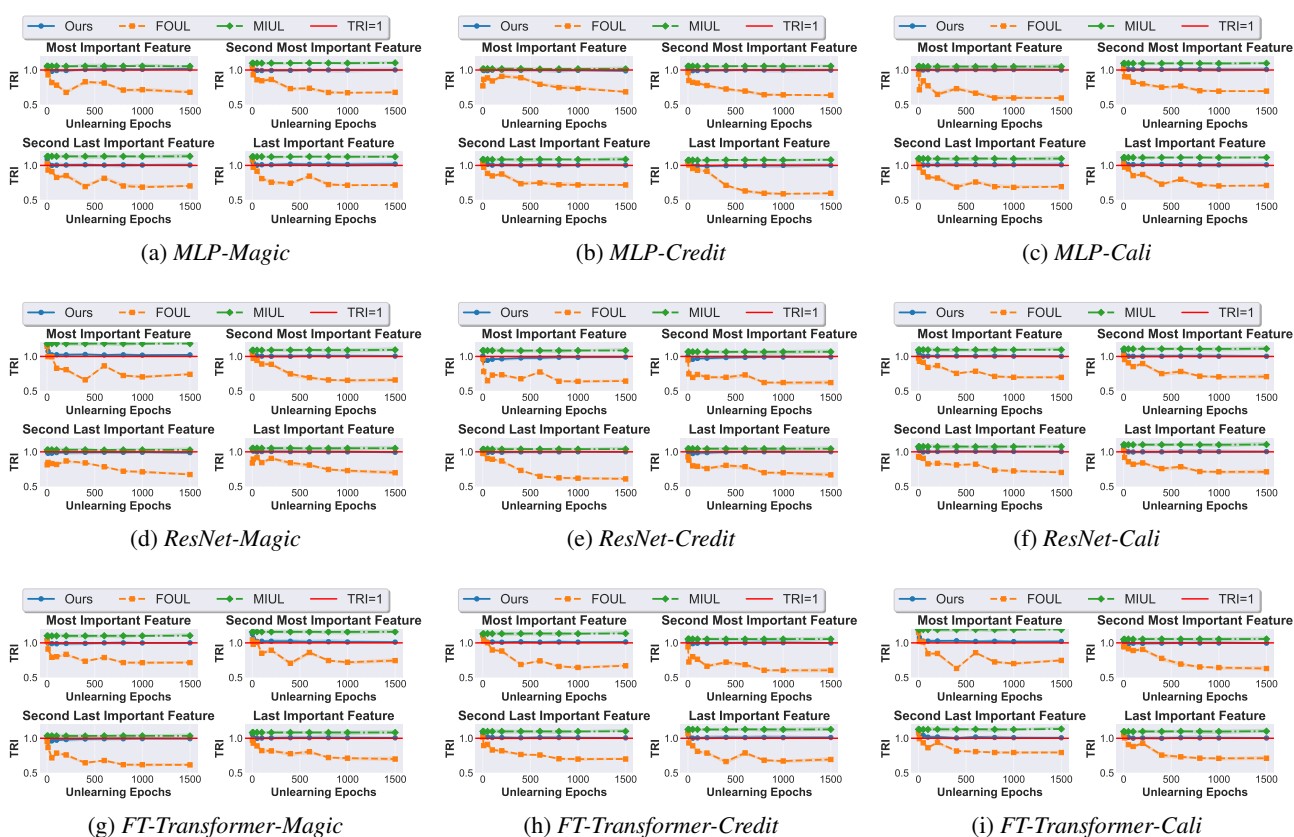

Figure 9: TRI results of unlearning features with different feature importance for tabular datasets.

Fig. 10 illustrates the EI results of unlearning the most, second most, second last, and last important features across the MLP, ResNet, and FT-Transformer models.

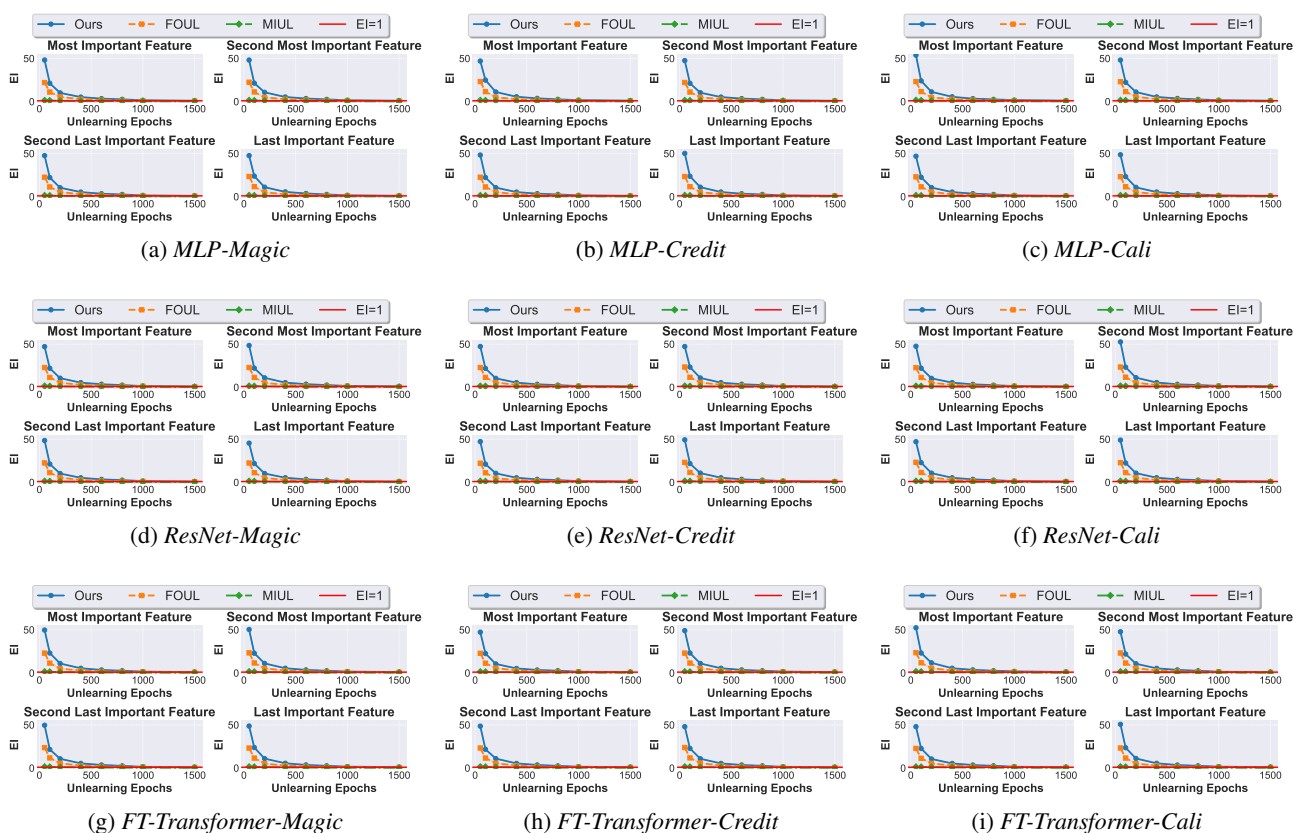

Figure 10: EI results of unlearning features with different feature importance for tabular datasets.

Fig. 11 illustrates the RASI results of unlearning the most, second most, second last, and last important features across the MLP, ResNet, and FT-Transformer models.

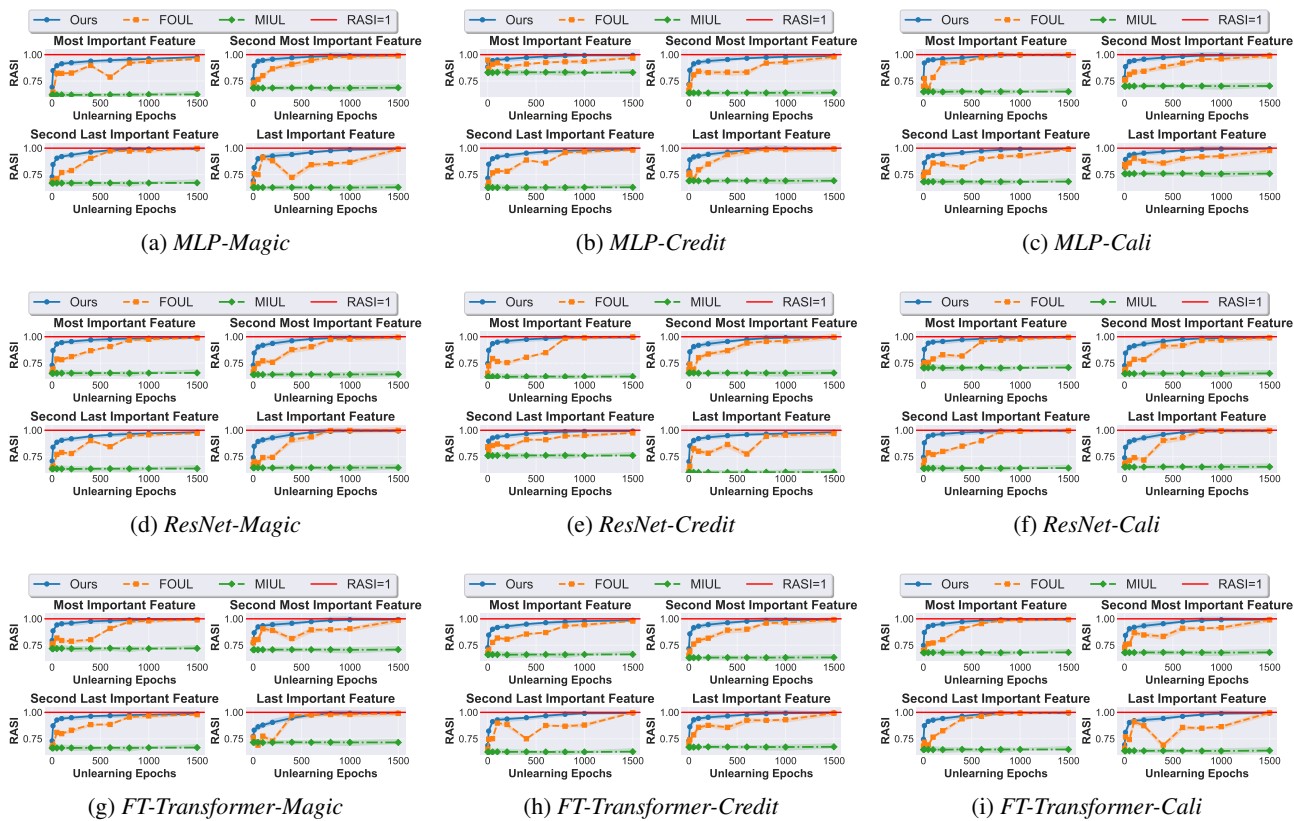

Figure 11: RASI results of unlearning features with different feature importance for tabular datasets.

Fig. 12 illustrates the SRI results of unlearning the most, second most, second last, and last important features across the MLP, ResNet, and FT-Transformer models.

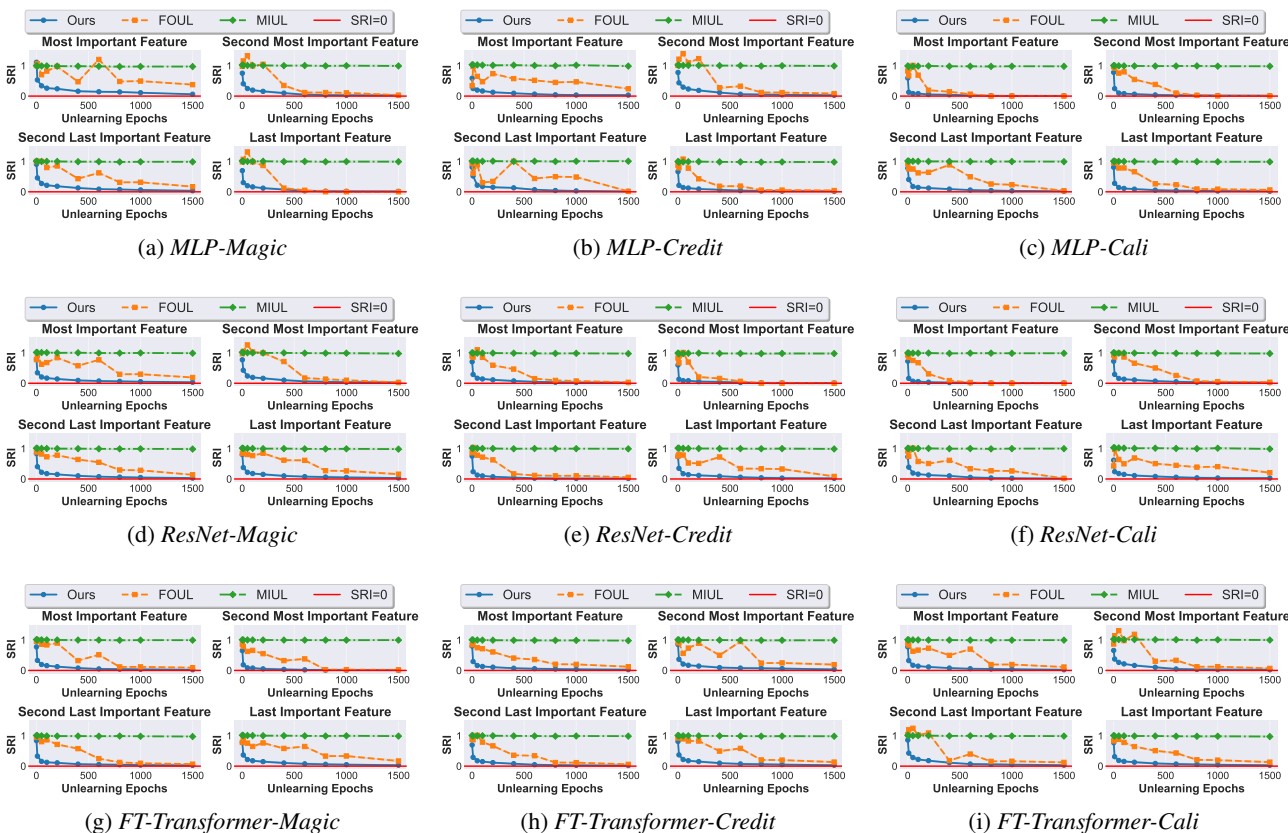

Figure 12: SRI results of unlearning features with different feature importance for tabular datasets.

Fig. 13 illustrates the SDI results of unlearning the most, second most, second last, and last important features across the MLP, ResNet, and FT-Transformer models.

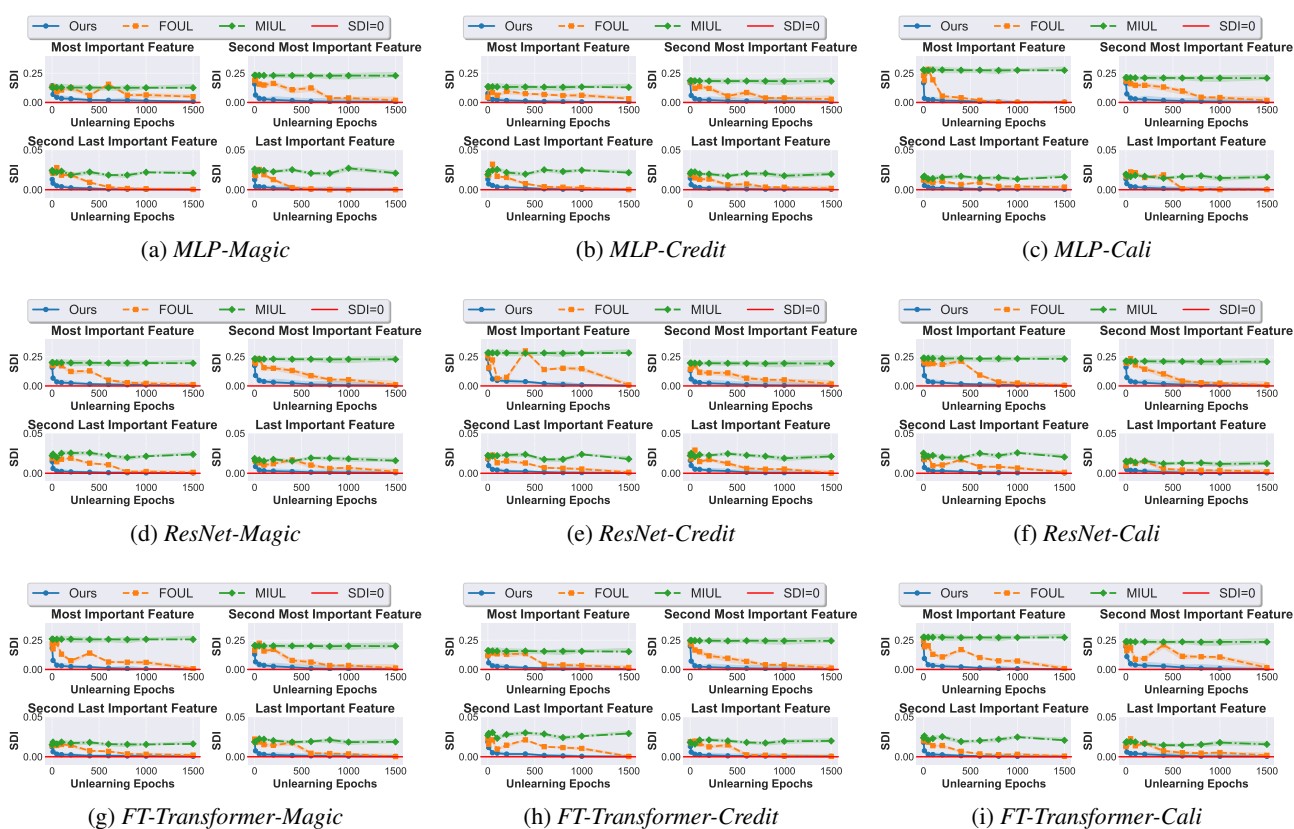

Figure 13: SDI results of unlearning features with different feature importance for tabular datasets.

## M  Feature Correlation Heatmaps of Used Tabular Datasets

In our experiments, we identify two features as highly-correlated features if the Pearson correlation coefficient is greater than 0.8.

Fig. 14 shows the feature correlation heatmap of the *Cali* dataset. The highly-correlated feature pairs of this dataset include `Latitude` and `Longitude`, `AveRooms`, and `AveBedrms`.

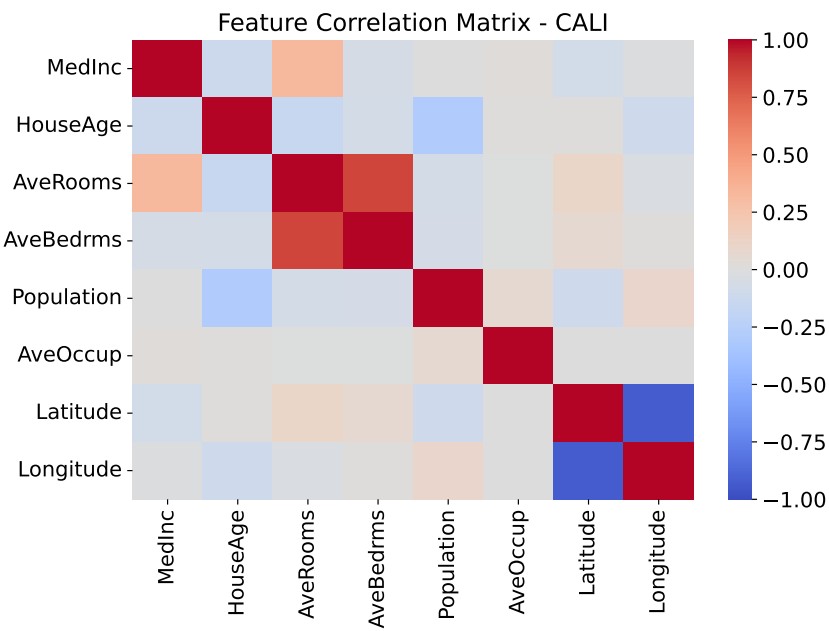

Figure 14: Feature correlation heatmap of the *Cali* dataset.

Fig. 15 shows the feature correlation heatmap of the *Credit* dataset. For readability, we use the abbreviations of each feature in the heatmap. The mapping between abbreviations and full names is provided in Table 7.

The highly-correlated feature pairs of this dataset include `NumberOfTimes90DaysLate`/`NumberOfTime60-89DaysPastDueNotWorse`, `NumberOfTime30-59DaysPastDueNotWorse`/`NumberOfTime60-89DaysPastDueNotWorse`, and `NumberOfTime30-59DaysPastDueNotWorse`/`NumberOfTimes90DaysLate`.

Table 7: Feature abbreviation in heatmap and full name in *Credit* dataset.

| Feature Full Name | Feature Abbreviation |
|---|---|
| `NumberOfTime30-59DaysPastDueNotWorse` | `No.30-59` |
| `NumberOfTimes90DaysLate` | `No.90` |
| `NumberOfTime60-89DaysPastDueNotWorse` | `No.60-89` |
| `NumberOfOpenCreditLinesAndLoans` | `No.Loan` |
| `age` | `age` |
| `RevolvingUtilizationOfUnsecuredLines` | `Rev.` |
| `NumberRealEstateLoansOrLines` | `No.Estate` |
| `MonthlyIncome` | `Income` |
| `NumberOfDependents` | `No.Dep` |
| `DebtRatio` | `Debt` |

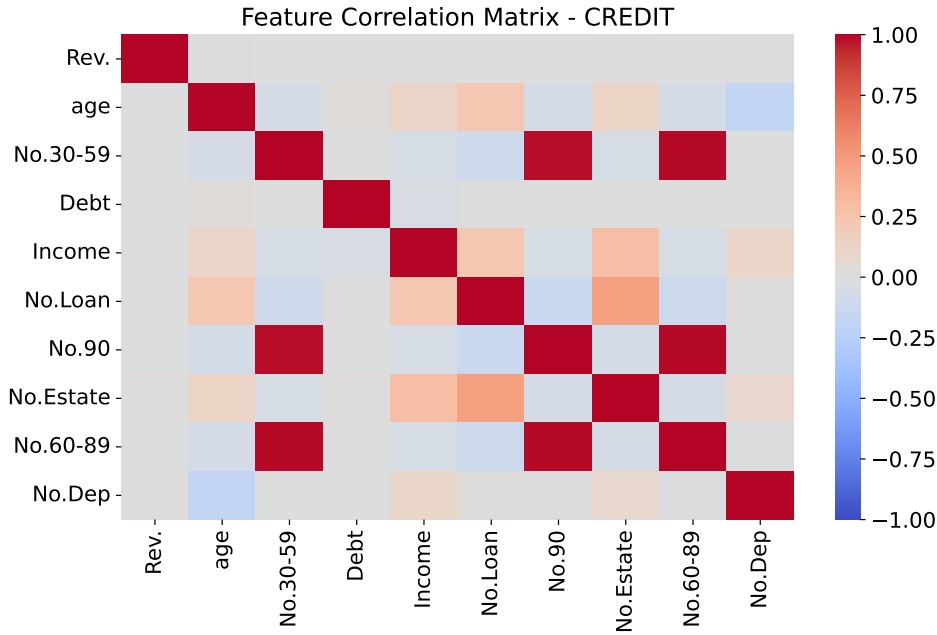

Figure 15: Feature correlation heatmap of the *Credit* dataset.

Fig. 16 shows the feature correlation heatmap of the *Magic* dataset. The highly-correlated feature pairs of this dataset include `fConc` and `fConc1`, `fSize` and `fConc`, `fSize` and `fConc1`.

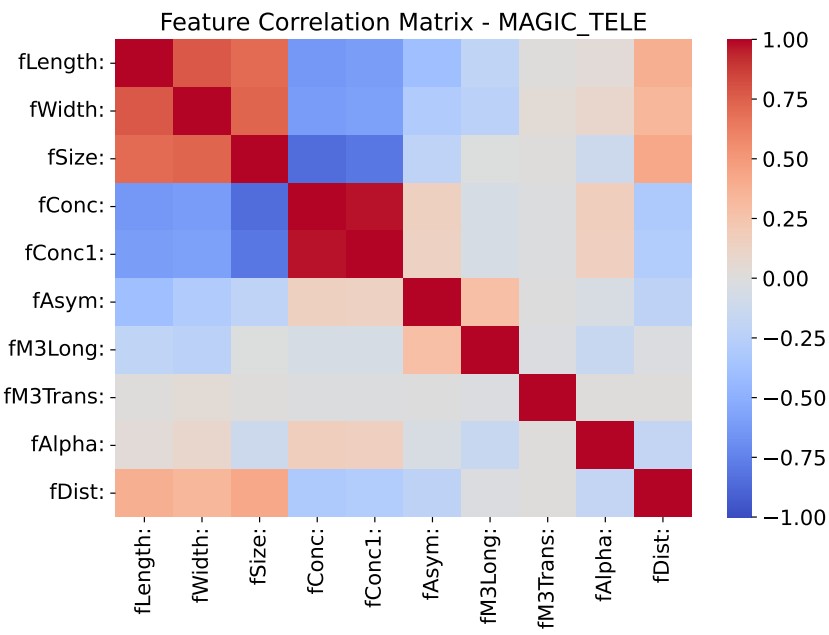

Figure 16: Feature correlation heatmap of the *Magic* dataset.

# N Full Results of Unlearning Features that are Highly-Correlated with Others

For the *Cali* dataset, we have identified two highly-correlated feature pairs in Appendix M – `Latitude` and `Longitude`, `AveRooms` and `AveBedrms`. Fig. 17, 18, and 19 show the evaluation results of unlearning `Latitude`, `Longitude`, and `AveBedrms`, respectively.

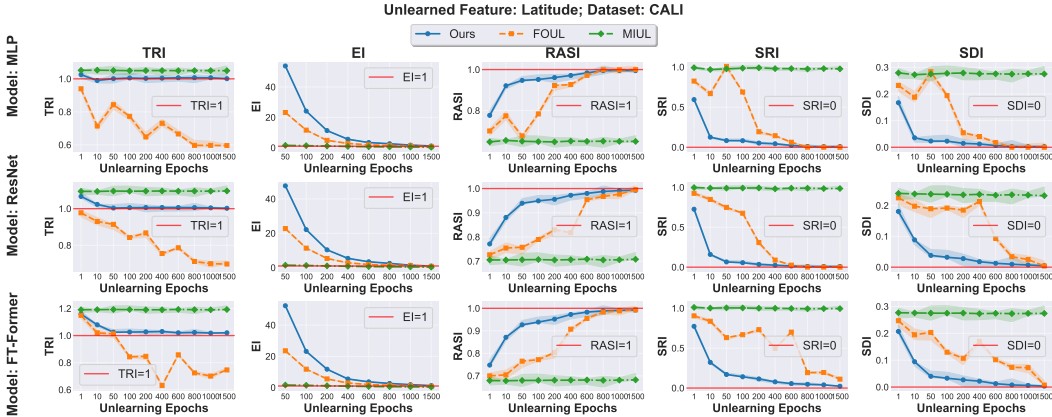

Figure 17: Evaluation results of unlearning `Latitude` of the *Cali* dataset.

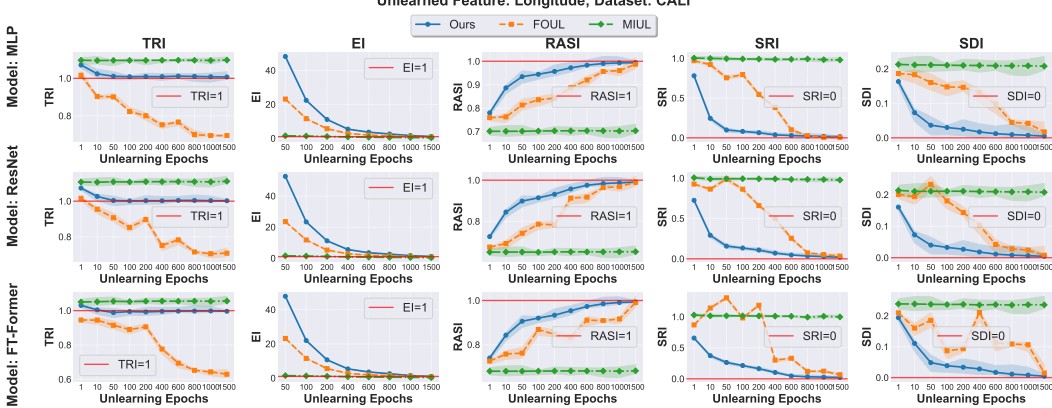

Figure 18: Evaluation results of unlearning `Longitude` of the *Cali* dataset.

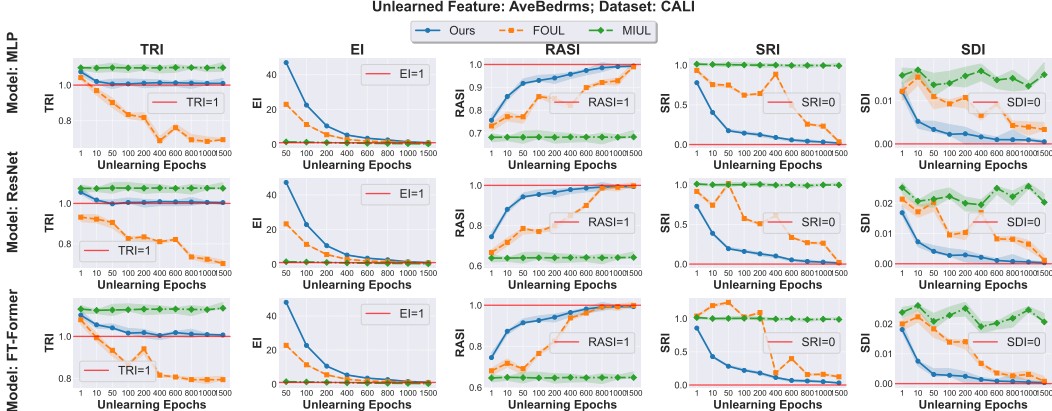

Figure 19: Evaluation results of unlearning `AveBedrms` of the *Cali* dataset.

For the *Credit* dataset, we have identified three highly-correlated feature pairs in Appendix M – `NumberOfTimes90DaysLate` and `NumberOfTime60-89DaysPastDueNotWorse`, `NumberOfTime30-59DaysPastDueNotWorse` and `NumberOfTime60-89DaysPastDueNotWorse`, `NumberOfTime30-59DaysPastDueNotWorse` and `NumberOfTimes90DaysLate`. Fig. 20 and 21 show the evaluation results of unlearning `NumberOfTime30-59DaysPastDueNotWorse` and `NumberOfTimes90DaysLate`, respectively.

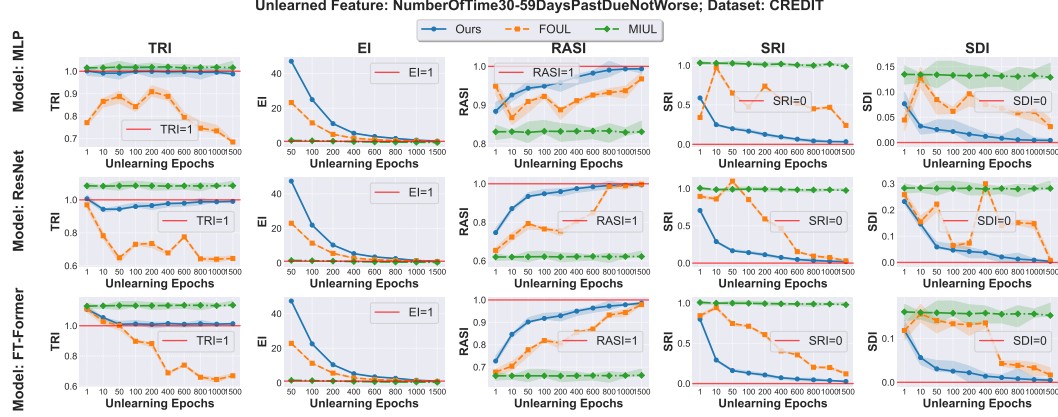

Figure 20: Evaluation results of unlearning `NumberOfTime30-59DaysPastDueNotWorse` of the *Credit* dataset.

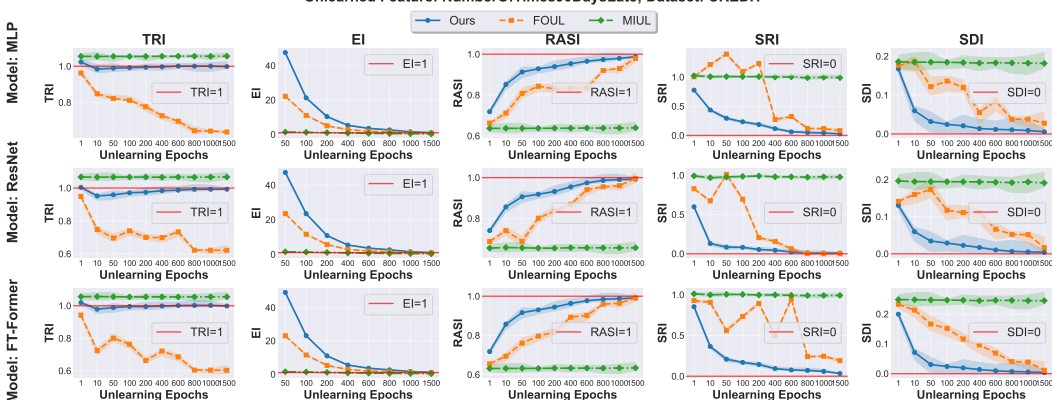

Figure 21: Evaluation results of unlearning `NumberOfTimes90DaysLate` of the *Credit* dataset.

For the *Magic* dataset, we have identified three highly-correlated feature pairs in Appendix M – `fConc` and `fConc1`, `fSize` and `fConc`, `fSize` and `fConc1`. Fig. 22 shows the evaluation results of unlearning `fSize`.

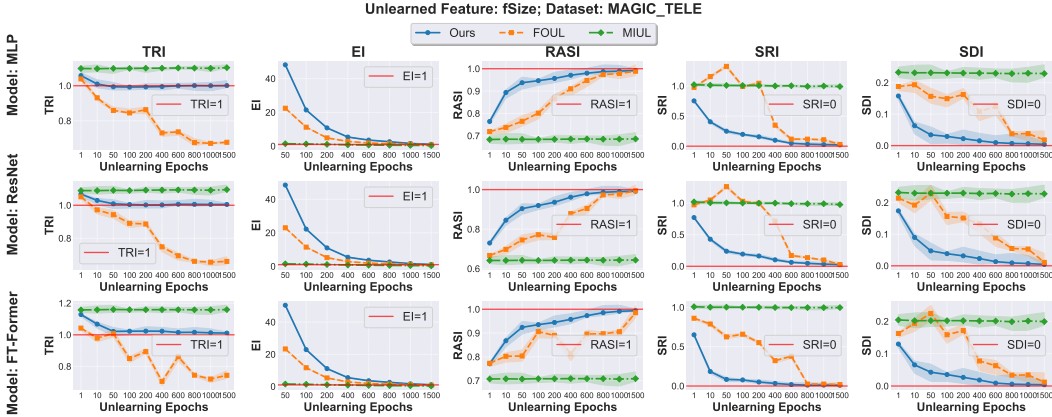

Figure 22: Evaluation results of unlearning `fSize` of the *Magic* dataset.

## O  Full Results of Fairness Evaluation Metrics under Single-Feature Unlearning Setting

All results shown below are averaged across all four unlearned features (i.e., the most, the second most, the second last, and the last important ones), all three neural architectures, and all used datasets. The features "HouseAge", "age", "fConc:" in the CALI, CREDIT, MAGIC_TELE datasets are considered sensitive attributes, with thresholds of $20$, $40$, $0.5$ to distinguish between privileged and unprivileged groups, respectively.

Table 8: Averaged DP difference comparison.

| Method \ Epoch | 1 | 10 | 50 | 100 | 200 | 400 | 600 | 800 | 1000 | 1500 |
|---|---|---|---|---|---|---|---|---|---|---|
| Ours | 0.0774 | 0.0330 | 0.0336 | 0.0127 | 0.0271 | 0.0147 | 0.0132 | 0.0206 | 0.0243 | 0.0097 |
| FOUL | 0.3791 | 0.2174 | 0.1654 | 0.3151 | 0.2065 | 0.0417 | 0.0598 | 0.1111 | 0.1054 | 0.2345 |
| MIUL | 0.1869 | 0.1319 | 0.0440 | 0.0954 | 0.0371 | 0.0424 | 0.0895 | 0.1026 | 0.0778 | 0.0291 |

Table 9: Averaged EO difference calculated via the true positive rate comparison.

| Method \ Epoch | 1 | 10 | 50 | 100 | 200 | 400 | 600 | 800 | 1000 | 1500 |
|---|---|---|---|---|---|---|---|---|---|---|
| Ours | 0.0394 | 0.0281 | 0.0310 | 0.0158 | 0.0288 | 0.0234 | 0.0175 | 0.0222 | 0.0192 | 0.0192 |
| FOUL | 0.1728 | 0.1463 | 0.2902 | 0.1616 | 0.2665 | 0.7684 | 0.5679 | 0.3727 | 0.1668 | 0.1290 |
| MIUL | 0.1841 | 0.2069 | 0.1477 | 0.0747 | 0.0719 | 0.0601 | 0.0574 | 0.0683 | 0.0958 | 0.1170 |

Table 10: Averaged EO difference calculated via the false positive rate comparison.

| Method \ Epoch | 1 | 10 | 50 | 100 | 200 | 400 | 600 | 800 | 1000 | 1500 |
|---|---|---|---|---|---|---|---|---|---|---|
| Ours | 0.1317 | 0.0462 | 0.0382 | 0.0156 | 0.0257 | 0.0160 | 0.0117 | 0.0207 | 0.0321 | 0.0134 |
| FOUL | 0.0752 | 0.0750 | 0.1627 | 0.1071 | 0.3891 | 0.6737 | 0.5946 | 0.3348 | 0.0927 | 0.1227 |
| MIUL | 0.1857 | 0.1249 | 0.0318 | 0.0782 | 0.0662 | 0.0557 | 0.1001 | 0.1252 | 0.0941 | 0.0218 |

# P    Full Results of Feature Unlearning for CV/Image Classification Tasks

Fig. 23 and 24 depict the results of unlearning visual features *nose* and *eyes* when performing classification for label class *Gender*.

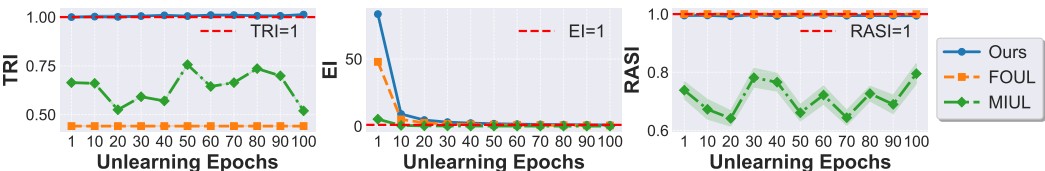

Figure 23: Evaluation results of unlearning *nose* for label class *Gender*.

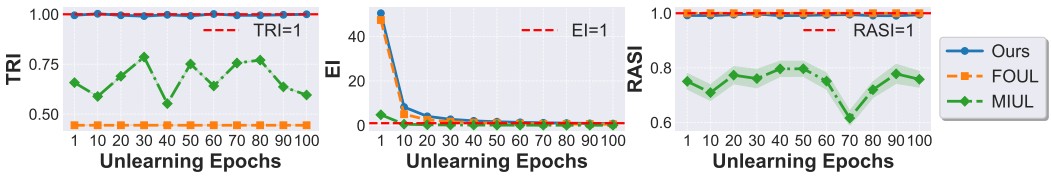

Figure 24: Evaluation results of unlearning *eyes* for label class *Gender*.

Fig. 25 depicts the results of unlearning visual features *nose* when performing classification for label class *Pointy Nose*.

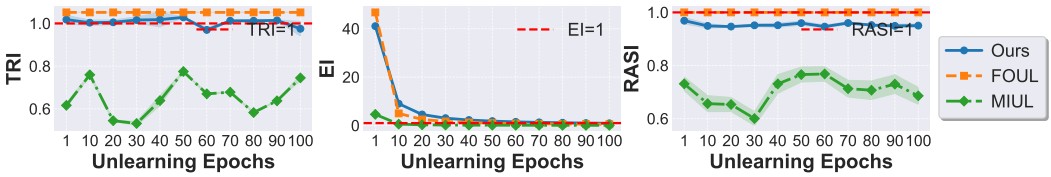

Figure 25: Evaluation results of unlearning *nose* for label class *Pointy Nose*.

Fig. 26 depicts the results of unlearning visual features *nose* when performing classification for label class *Big Nose*.

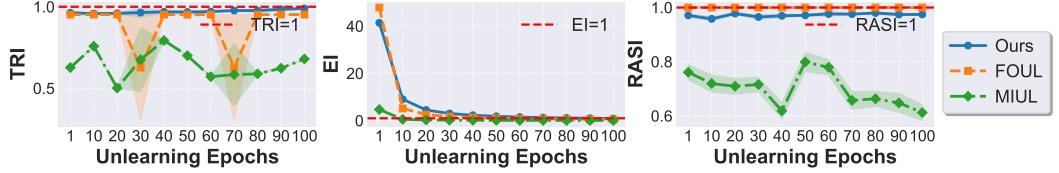

Figure 26: Evaluation results of unlearning *nose* for label class *Big Nose*.

Fig. 27 depicts the results of unlearning visual features *eyes* when performing classification for label class *Narrow Eyes*.

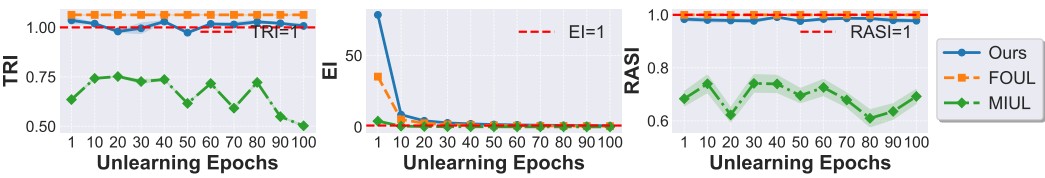

Figure 27: Evaluation results of unlearning *eyes* for label class *Narrow Eyes*.

# Q Full Experimental Results of Multi-Feature Unlearning for Tabular Dataset

The following figures from Fig. 29 to Fig. 30 depict evaluation results w.r.t. defined all five evaluation metrics across all used datasets for multi-feature unlearning, with results presented in from Table 11 to Table 13 for cross comparison.

Table 11: Comparison of unlearning evaluation results on *Comp* dataset. The terms "Eva.Me." and "Algo." are abbreviations for "Evaluation Metrics" and "Algorithm", respectively.

| Eva.Me. | Algo. | Unlearning Epochs | | | | | | | | | |
|---|---|---|---|---|---|---|---|---|---|---|---|
| | | 1 | 10 | 50 | 100 | 200 | 400 | 600 | 800 | 1000 | 1500 |
| *TRI* | *FOUL* | $108\% \pm 0\%$ | $93\% \pm 0\%$ | $94\% \pm 0\%$ | $85\% \pm 0\%$ | $78\% \pm 0\%$ | $75\% \pm 0\%$ | $56\% \pm 0\%$ | $66\% \pm 0\%$ | $69\% \pm 0\%$ | $61\% \pm 0\%$ |
| | *MIUL* | $108\% \pm 0.2\%$ | $108\% \pm 0.2\%$ | $108\% \pm 0.3\%$ | $108\% \pm 0.4\%$ | $108\% \pm 0.2\%$ | $108\% \pm 0.4\%$ | $109\% \pm 0.2\%$ | $108 \pm 0.3\%$ | $108\% \pm 0.5\%$ | $108\% \pm 0.4\%$ |
| | Ours | $\mathbf{105\% \pm 1.1\%}$ | $\mathbf{97\% \pm 0.6\%}$ | $\mathbf{94\% \pm 1.5\%}$ | $\mathbf{93\% \pm 0.8\%}$ | $\mathbf{95\% \pm 0.7\%}$ | $\mathbf{96\% \pm 0.3\%}$ | $\mathbf{97\% \pm 0.3\%}$ | $\mathbf{97\% \pm 0.4\%}$ | $\mathbf{98\% \pm 0.8\%}$ | $\mathbf{98\% \pm 0.4\%}$ |
| *EI* | *FOUL* | 860 | 87 | 22 | 11 | 5.6 | 2.8 | 1.9 | 1.4 | 1.0 | 0.7 |
| | *MIUL* | 1.7 | 1.6 | 1.5 | 1.3 | 1 | 0.8 | 0.6 | 0.5 | 0.4 | 0.3 |
| | Ours | **1270** | **170** | **39** | **19** | **9.7** | **4.9** | **3.1** | **2.5** | **2.0** | **1.3** |
| *RASI* | *FOUL* | $58\% \pm 0\%$ | $68\% \pm 0.1\%$ | $67\% \pm 0.2\%$ | $69\% \pm 0.1\%$ | $63\% \pm 0.1\%$ | $68\% \pm 0.3\%$ | $62\% \pm 0.2\%$ | $77\% \pm 0.1\%$ | $67\% \pm 0.1\%$ | $79\% \pm 0\%$ |
| | *MIUL* | $57\% \pm 0.3\%$ | $57\% \pm 0.6\%$ | $57\% \pm 0.1\%$ | $57\% \pm 0.1\%$ | $57\% \pm 0.3\%$ | $57\% \pm 0.6\%$ | $57\% \pm 0.3\%$ | $57\% \pm 0.1\%$ | $57\% \pm 0.8\%$ | $57\% \pm 0.3\%$ |
| | Ours | $\mathbf{61\% \pm 1.6\%}$ | $\mathbf{85\% \pm 0.8\%}$ | $\mathbf{87\% \pm 1.3\%}$ | $\mathbf{90\% \pm 2.6\%}$ | $\mathbf{91\% \pm 1.5\%}$ | $\mathbf{94\% \pm 0.8\%}$ | $\mathbf{96\% \pm 0.1\%}$ | $96\% \pm 0.1\%$ | $97\% \pm 0.4\%$ | $98\% \pm 0.3\%$ |
| *SRI* | *FOUL* | $96\% \pm 3.0\%$ | $65\% \pm 3.0\%$ | $154\% \pm 2.2\%$ | $221\% \pm 0.7\%$ | $247\% \pm 2.9\%$ | $230\% \pm 0.2\%$ | $31\% \pm 0.8\%$ | $65\% \pm 2.4\%$ | $199\% \pm 0.6\%$ | $22\% \pm 0.5\%$ |
| | *MIUL* | $110\% \pm 3.7\%$ | $95\% \pm 5.6\%$ | $109\% \pm 6.4\%$ | $100\% \pm 0.7\%$ | $101\% \pm 1.1\%$ | $101\% \pm 8.2\%$ | $110\% \pm 10\%$ | $112\% \pm 3.1\%$ | $102\% \pm 1.3\%$ | $100\% \pm 7.4\%$ |
| | Ours | $\mathbf{81\% \pm 8.8\%}$ | $\mathbf{51\% \pm 13.8\%}$ | $\mathbf{58\% \pm 21.2\%}$ | $\mathbf{52\% \pm 9.1\%}$ | $\mathbf{26\% \pm 3.0\%}$ | $\mathbf{19\% \pm 2.1\%}$ | $\mathbf{12\% \pm 1.5\%}$ | $\mathbf{11\% \pm 1.8\%}$ | $\mathbf{12\% \pm 4.8\%}$ | $\mathbf{8.0\% \pm 1.6\%}$ |
| *SDI* | *FOUL* | $7.0\% \pm 0.2\%$ | $5.0\% \pm 0\%$ | $11\% \pm 0.2\%$ | $16\% \pm 0.1\%$ | $18\% \pm 0.2\%$ | $16\% \pm 0.2\%$ | $2.0\% \pm 0.1\%$ | $5.0\% \pm 2.0\%$ | $14\% \pm 0\%$ | $2.0\% \pm 0\%$ |
| | *MIUL* | $8.0\% \pm 0.3\%$ | $7.0\% \pm 0.4\%$ | $8.0\% \pm 0.5\%$ | $7.0\% \pm 0.1\%$ | $7.0\% \pm 0.1\%$ | $7.0\% \pm 0.6\%$ | $8.0\% \pm 7.0\%$ | $8.0\% \pm 0.2\%$ | $7.0\% \pm 0.1\%$ | $7.0\% \pm 0.5\%$ |
| | Ours | $\mathbf{6.0\% \pm 0.6\%}$ | $\mathbf{4.0\% \pm 1.0\%}$ | $\mathbf{4.0\% \pm 1.5\%}$ | $\mathbf{4.0\% \pm 0.7\%}$ | $\mathbf{2.0\% \pm 0.2\%}$ | $\mathbf{1.0\% \pm 0.1\%}$ | $\mathbf{1.0\% \pm 0.1\%}$ | $\mathbf{1.0\% \pm 0.1\%}$ | $\mathbf{1.0\% \pm 0.3\%}$ | $\mathbf{1.0\% \pm 0.1\%}$ |

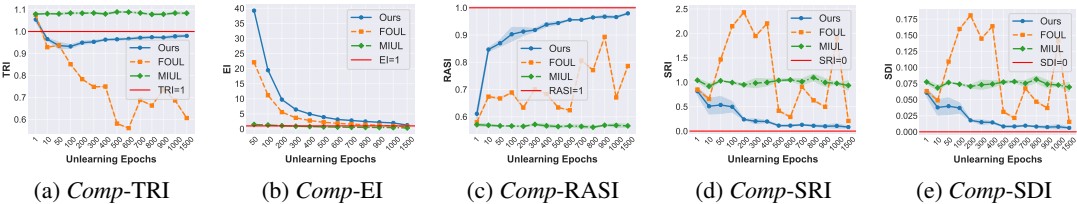

(a) *Comp*-TRI     (b) *Comp*-EI     (c) *Comp*-RASI     (d) *Comp*-SRI     (e) *Comp*-SDI

Figure 28: Unlearning evaluation results of the *Comp* dataset under the multi-feature unlearning setting.

Table 12: Comparison of unlearning evaluation results on *Pol* dataset. The terms "Eva.Me." and "Algo." are abbreviations for "Evaluation Metrics" and "Algorithm", respectively.

| Eva.Me. | Algo. | Unlearning Epochs | | | | | | | | | |
|---|---|---|---|---|---|---|---|---|---|---|---|
| | | 1 | 10 | 50 | 100 | 200 | 400 | 600 | 800 | 1000 | 1500 |
| *TRI* | *FOUL* | 179%±0% | 177%±0% | 159%±0% | 160%±0% | 142%±0% | 126%±0% | 145%±0% | 158%±0% | 160%±0% | 164%±0% |
| | *MIUL* | 181%±0.2% | 180%±0.2% | 181%±0.6% | 181%±0.2% | 181%±0.3% | 181%±0.4% | 181%±0.2% | 181±0.1% | 181%±0.1% | 181%±0.3% |
| | Ours | **179%±0.6%** | **175%±2.5%** | **146%±25.2%** | **110%±0.6%** | **105%±3.5%** | **102%±1.0%** | **105%±2.7%** | **105%±4.6%** | **102%±0.4%** | **106%±4.3%** |
| *EI* | *FOUL* | 736 | 106 | 22 | 10 | 5.5 | 2.8 | 1.9 | 1.4 | 1.1 | 0.7 |
| | *MIUL* | 1.6 | 1.5 | 1.4 | 1.2 | 1.0 | 0.8 | 0.6 | 0.5 | 0.4 | 0.3 |
| | Ours | **1032** | **183** | **34** | **17** | **9.8** | **4.9** | **3.1** | **2.3** | **1.9** | **1.3** |
| *RASI* | *FOUL* | 53%±0.3% | 52%±0.2% | 55%±0.2% | 55%±0.1% | 64%±0.1% | 71%±0.2% | 61%±0.1% | 56%±0.4% | 56%±0.2% | 55%±0.1% |
| | *MIUL* | 52%±0.3% | 52%±0.2% | 52%±0.4% | 52%±0.4% | 52%±0.4% | 52%±0.3% | 52%±0.4% | 52%±0.6% | 52%±0.3% | 52%±0.3% |
| | Ours | **53%±0.3%** | **53%±0.1%** | **65%±16%** | **89%±1.7%** | **96%±3.6%** | **98%±2.9%** | **96%±2.8%** | **96%±4.7%** | **99%±0.4%** | **99%±0.4%** |
| *SRI* | *FOUL* | 101%±0.8% | 100%±0.5% | 91%±0.1% | 96%±1.8% | 72%±0.8% | 47%±0.5% | 46%±1.5% | 65%±0.8% | 69%±1.9% | 72%±0.4% |
| | *MIUL* | 101%±1.5% | 98%±0.8% | 99%±1.2% | 98%±0.5% | 96%±1.1% | 98%±3.5% | 94%±3.0% | 95%±1.7% | 91%±4.1% | 72%±0.4% |
| | Ours | **99%±2.5%** | **93%±3.6%** | **55%±3.7%** | **4.0%±2.0%** | **3.0%±1.8%** | **3.0%±3.0%** | **5.0%±2.1%** | **3.0%±2.7%** | **3.0%±0.2%** | **3.0%±1.3%** |
| *SDI* | *FOUL* | 16%±0.1% | 15%±0.1% | 14%±0% | 15%±0.3% | 11%±0.1% | 7.0%±0.2% | 7.0%±0.2% | 10%±0.1% | 11%±0.3% | 11%±0.1% |
| | *MIUL* | 16%±0.2% | 15%±0.1% | 15%±0.2% | 15%±0.1% | 15%±0.2% | 15%±0.5% | 14%±0.5% | 15%±0.1% | 14%±0.6% | 14%±0.6% |
| | Ours | **15%±0.4%** | **14%±0.6%** | **8.0%±5.7%** | **1.0%±0.3%** | **1.0%±0.3%** | **0%±0.5%** | **1.0%±0.3%** | **0%±0.4%** | **0%±0%** | **0%±0.2%** |

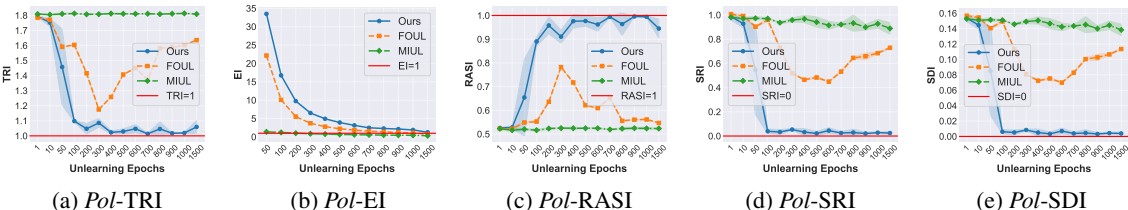

(a) *Pol*-TRI  (b) *Pol*-EI  (c) *Pol*-RASI  (d) *Pol*-SRI  (e) *Pol*-SDI

Figure 29: Unlearning evaluation results of the *Pol* dataset.

Table 13: Comparison of unlearning evaluation results on *Eye* dataset. The terms "Eva.Me." and "Algo." are abbreviations for "Evaluation Metrics" and "Algorithm", respectively.

| Eva.Me. | Algo. | Unlearning Epochs | | | | | | | | | |
|---|---|---|---|---|---|---|---|---|---|---|---|
| | | 1 | 10 | 50 | 100 | 200 | 400 | 600 | 800 | 1000 | 1500 |
| *TRI* | *FOUL* | 103%±0% | 92%±0% | 86%±0% | 90%±0% | 85%±0% | 86%±0% | 88%±0% | 86%±0% | 86%±0% | 86%±0% |
| | *MIUL* | 105%±0.7% | 105%±0.4% | 106%±0.5% | 106%±1.1% | 107%±0.1% | 105%±0.9% | 105%±0.6% | 106±0.6% | 105%±0.7% | 105%±1.8% |
| | Ours | **102%±0.1%** | **101%±1.6%** | **100%±1.8%** | **100%±0.6%** | **103%±0.6%** | **100%±1.3%** | **102%±2.8%** | **101%±1.7%** | **102%±2.3%** | **102%±1.4%** |
| *EI* | *FOUL* | 680 | 105 | 17 | 11 | 5.5 | 2.8 | 1.8 | 1.4 | 1.1 | 0.7 |
| | *MIUL* | 1.5 | 1.5 | 1.4 | 1.2 | 1 | 0.8 | 0.6 | 0.5 | 0.4 | 0.3 |
| | Ours | **871** | **178** | **38** | **19** | **9.7** | **4.2** | **3.2** | **2.3** | **1.9** | **1.3** |
| *RASI* | *FOUL* | 55%±0.4% | 71%±0.4% | **83%±0%** | **91%±0.1%** | 60%±0.5% | **99%±0%** | 78%±0.2% | **99%±0%** | **99%±0%** | **100%±0%** |
| | *MIUL* | 57%±0.3% | 56%±0.2% | 57%±0.3% | 56%±0.5% | 56%±0.2% | 56%±0.9% | 57%±0.5% | 57%±0.3% | 57%±0.3% | 57%±0.4% |
| | Ours | **59%±1.4%** | **73%±3.3%** | 79%±2.6% | 79%±2.6% | **81%±3.3%** | 86%±2.3% | **87%±2.0%** | 84%±1.7% | 89%±0.9% | 90%±0.9% |
| *SRI* | *FOUL* | **79%±1.8%** | **49%±2.0%** | **45%±1.5%** | **15%±0.5%** | 174%±2.0% | **1.0%±0.2%** | 56%±0.6% | **1.0%±0.3%** | **1.0%±0.1%** | **0%±0%** |
| | *MIUL* | 93%±6.6% | 95%±5.0% | 97%±3.4% | 97%±4.2% | 86%±2.4% | 87%±4.1% | 91%±4.2% | 91%±4.5% | 92%±5.1% | 92%±1.9% |
| | Ours | 104%±18% | 87%±23% | 48%±12% | 80%±28% | **32%±11%** | 19%±2.4% | **34%±18.5%** | 19%±4.3% | 25%±7.2% | 29%±5% |
| *SDI* | *FOUL* | **3.0%±0.1%** | **2.0%±0.1%** | 2.0%±0.1% | **1.0%±0%** | 7.0%±0.1% | **0%±0%** | 2.0%±0% | **0%±0%** | **0%±0%** | **0%±0%** |
| | *MIUL* | 4.0%±0.3% | 4.0%±0.2% | 4.0%±0.1% | 4.0%±0.2% | 4.0%±0.1% | 4.0%±0.2% | 4.0%±0.2% | 4.0%±0.2% | 4.0%±0.2% | 4.0%±0.1% |
| | Ours | 4.0%±0.8% | 4.0%±0.9% | **2.0%±0.5%** | 3.0%±1.2% | **1.0%±0.5%** | 1.0%±0.1% | **1.0%±0.8%** | **1.0%±0.2%** | 1.0%±0.3% | 1.0%±0.2% |

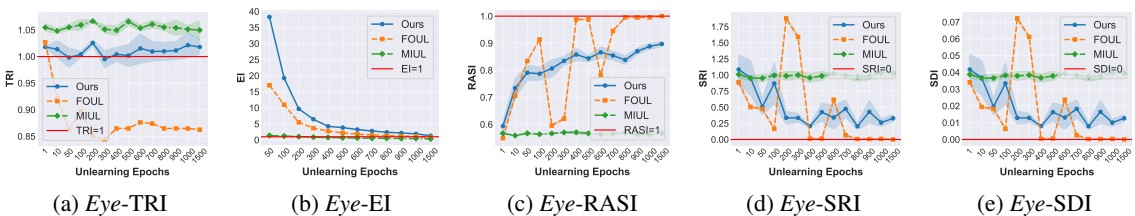

(a) *Eye*-TRI  (b) *Eye*-EI  (c) *Eye*-RASI  (d) *Eye*-SRI  (e) *Eye*-SDI

Figure 30: Unlearning evaluation results of the *Eye* dataset.

