# OpenReview forum: "Feature Unlearning: Theoretical Foundations and Practical Applications with Shuffling"
_NeurIPS.cc/2025/Conference — NeurIPS 2025 poster_

### Official Review · Reviewer_nUD2 · 2025-06-03

**Clarity:** 3
**Significance:** 3
**Originality:** 3
**Rating:** 5
**Confidence:** 2

**Summary:**

This paper introduces a simple approach for feature unlearning in tabular and image data which is backed up theoretically and evaluated with extensive experiments.

**Questions:**

Why did the authors leave out bias/discrimination/privacy metrics from the evaluation? (main reason for the currently lower score; I am happy to increase my score if this is addressed (or if I misunderstood something))

**Ethical Concerns:**

["NO or VERY MINOR ethics concerns only"]

**Final Justification:**

The authors have addressed my remaining concerns (additional experiments to show MIA results that were not present in the original paper) in the rebuttal. I have therefore increased my score to accept.

**Limitations:**

yes

**Quality:**

2

**Strengths And Weaknesses:**

Strengths:
- Simple and elegant method with theoretical backing
- Extensive experimentation across datasets, model architectures, and modalities
- Original and well-written paper

Weaknesses:
- Lack of citations in 5.1 - where does TRI, RASI, etc. come from?
- The main weakness of this paper is that the authors do not investigate if an attacker can infer if the feature was used during the original training or not. This is a typical metric (e.g. via Membership Inference Attacks in class unlearning) for unlearning, as accuracy alone often does not tell the full picture. The authors state “Detecting privacy leaks in learning
models, which is a complex issue, falls outside the scope of this research.” Given that Figure 1 uses clinical data as an example, shouldn’t data privacy be discussed? Directly linked to this, if we can detect that the model was trained with a feature, a bias may still remain from that feature that was not properly removed. For this I would have expected metrics from related fields (bias & discrimination) in this paper. The practical significance in unlearning features lies in privacy and bias, therefore this is a weakness of this paper.

---

> ### Author Rebuttal · Authors · 2025-07-26
>
> We thank the reviewer for their feedback and for engaging with our work. We have carefully considered the comments and revised our manuscript/clarified our responses accordingly.
>
> **Response to “Lack of citations in 5.1 - where does TRI, RASI, etc. come from”**
>
> Thank you for your suggestions on adding references to the evaluation metrics.
>
> The metrics Test Retention Index (TRI), Efficiency Index (EI) are stemmed from the reference [1], which was accepted by EuroS&P 2025.
>
> Our study originally designed the metric Unlearning Robustness Against Shuffling Index (RASI) based on the shuffling characteristic of our shuffling-based approach. To calculate this metric, we shuffled the values of unlearned features within the test set and then observed whether such shuffling affects prediction outcomes. As a result, RASI can examine whether the unlearned model has successfully eliminated dependency on the unlearned feature regarding model predictions.
>
> The other two metrics, i.e., SHAP Retention Index (SRI) and SHAP Distance-to-zero Index (SDI), were inspired by explainable AI techniques. In our study, we calculate each feature’s importance using the SHapley Additive exPlanations (SHAP). By applying an effective unlearning algorithm, the unlearned feature’s importance should be reduced to zero. The designed two SHAP-related metrics are able to investigate the targeted feature’s importance after unlearning.
>
> [1] Xavier F. Cadet, Anastasia Borovykh, Mohammad Malekzadeh, Sara Ahmadi-Abhari, and Hamed Haddadi. Deep unlearn: Benchmarking machine unlearning
>
> **Response to “Why did the authors leave out bias/discrimination/privacy metrics from the evaluation”**
>
> We thank the reviewer for insightful feedback. We agree that evaluating the practical significance of feature unlearning requires auditing for both privacy and fairness, as accuracy alone is insufficient.
>
> In our revised manuscript, we now include 1) a feature-level membership inference attack (MIA) and 2) fairness audits to evaluate our method in terms of privacy and fairness, respectively.
>
> For 1), we adapt the standard MIA to the feature level. Instead of asking “was this individual data point in the training dataset?”, the feature-level MIA asks “was this feature used to train the model?”. For a successful unlearning algorithm, a feature-level attacker cannot distinguish the unlearned model from a model that is retrained from scratch without the unlearned feature.
>
> We implement the feature-level MIA in the following steps. Firstly, we create an attack dataset. For each sample, we query $f_{\theta’}$ (model after unlearning) and $g_\phi$ (retrained-from-scratch model) to get their outputs, i.e., the full vector of prediction probabilities. Then, we label the outputs from $f_{\theta’}$ as class “Unlearned” and the outputs from $g_\phi$ as class “Retrained”.
>
> Subsequently, we train an attack model – a RandomForecast binary classifier (via the $\texttt{sklearn}$ Python library) on this built attack dataset. This attacker’s job is to learn the subtle differences between the outputs distributions of our unlearned model ($f_{\theta’}$) and a truly “clean” model ($g_\phi$).
>
> Finally, we evaluate this trained attack model on a holdout set of queries. The final “Attack Accuracy” is the new metric regarding privacy. **An accuracy near 50% MIA suggests that the attacker cannot do better than random guessing. Our unlearned model ($f_{\theta’}$) is indistinguishable from a model retrained from scratch ($g_\phi$), which indicates that our unlearning is successful from the privacy perspective.**
>
>
> For 2), we introduce two standard fairness metrics – demographic parity (DP) and equalized odds (EO). **For a successful unlearning, the fairness score of the unlearned model should be very close to the retrained-from-scratch model.**
>
> The DP score is calculated in the following steps. Firstly, we identify several key components related to DP, including
> - Unprivileged/Privileged Groups: The group that has historically faced a disadvantage/advantage (e.g., for features like “Race” or “Gender”).
> - Favorable Outcome: The desired model prediction (e.g., “loan approved”).
> - Model Predictions: The output labels generated by our unlearned model or the retrained-from-scratch model.
>
> Next, for both the privileged and unprivileged groups, we calculate the rate at which they receive the favorable outcome from models. The formula for the rate is calculated as
> $$
> \textrm{Rate} = \frac{\textrm{Number of individuals in the group who received the favorable outcome​}}{\textrm{Total number of individuals in the group}}
> $$
>
> The above rates are denoted as $\textrm{DP(Label=Favorable Outcome | Privileged Group)}$ and $\textrm{DP(Label=Favorable Outcome | Unprivileged Group)}$, respectively.
>
> Finally, we calculate the DP differences between unprivileged and privileged groups, formulated as
> $$
> \textrm{DP Difference} = \textrm{DP(Label=Favorable Outcome | Unprivileged Group) - \textrm{DP(Label=Favorable Outcome | Privileged Group)}}
> $$
>
> **A DP difference of zero indicates perfect fairness.**
>
> For the EO difference, we calculate the differences of True Positive Rate (TPR), False Positive Rate (FPR) between the privileged and the unprivileged groups.
> $$\textrm{EO-TPR Difference} = \Delta\textrm{TPR}$$
> $$\textrm{EO-FPR Difference} = \Delta\textrm{FPR}$$
>
> **Both metrics should be as close as to zero from the perspective of fairness.**
>
> In the following, we present the results of feature-level MIA attack accuracy, DP difference, and EO differences (including both TPR and FPR) across all three adopted neural architectures (i.e., MLP, ResNet, and FT-Former). All results shown below are averaged across all four unlearned features, all three neural architectures, and all used datasets. In particular, the features “HouseAge”, “age”, “fConc:” in the *CALI*, *CREDIT*, *MAGIC_TELE* datasets are considered sensitive attributes, with thresholds of 20, 40, 0.5 to distinguish between privileged and unprivileged groups, respectively.
>
> **The below results demonstrate the effectiveness of our unlearning methods in terms of privacy and fairness.**
>
> ***Table 1**: Averaged MIA attack accuracy across our unlearning method and the two baselines (optimal: 0.5)*
> | Method | Epoch 1 | Epoch 10 | Epoch 50 | Epoch 100 | Epoch 200 | Epoch 400 | Epoch 600 | Epoch 800 | Epoch 1000 | Epoch 1500 |
> |:-------|:----------|:----------|:----------|:----------|:----------|:----------|:----------|:----------|:----------|:-----------|
> | **Ours**   |     0.5437      |      0.5204     |   0.4964        |      0.5080     |      0.5254     |  0.5278 |   0.5246      |    0.5202       |      0.5121     |   0.5111        |
> | FOUL   |      0.5736     |    0.6197       |    0.6029       |     0.5879      |      0.5690     |      0.6512     |     0.7512      |    0.7630       |    0.7630       |      0.7630      |
> | MIUL   |     0.6509      |     0.5963      |    0.5973       |    0.5966       |    0.6049       |  0.6029 |  0.5950       |     0.6120      |   0.5938        |           0.6139 |
>
>
> ***Table 2**: Averaged DP difference across our unlearning method and the two baselines (optimal: 0)*
> | Method | Epoch 1 | Epoch 10 | Epoch 50 | Epoch 100 | Epoch 200 | Epoch 400 | Epoch 600 | Epoch 800 | Epoch 1000 | Epoch 1500 |
> |:-------|:----------|:----------|:----------|:----------|:----------|:----------|:----------|:----------|:----------|:-----------|
> | **Ours**   |     0.0774      |    0.0330       |      0.0336     |     0.0127      |    0.0271       |    0.0147       |     0.0132      |    0.0206       |     0.0243      |        0.0097    |
> | FOUL   |      0.3791     |     0.2174      |   0.1654        |    0.3151       |    0.2065       |      0.0417     |    0.0598       |    0.1111       |   0.1054        |    0.2345        |
> | MIUL   |     0.1869      |    0.1319       |     0.0440      |     0.0954      |    0.0371       |    0.0424       |     0.0895      |     0.1026      |      0.0778     |     0.0291       |
>
>
> ***Table 3**: Averaged EO-TPR difference across our unlearning method and the two baselines (optimal: 0)*
> | Method | Epoch 1 | Epoch 10 | Epoch 50 | Epoch 100 | Epoch 200 | Epoch 400 | Epoch 600 | Epoch 800 | Epoch 1000 | Epoch 1500 |
> |:-------|:----------|:----------|:----------|:----------|:----------|:----------|:----------|:----------|:----------|:-----------|
> | **Ours**   |     0.0394      |    0.0281       |    0.0310       |      0.0158     |     0.0288      |    0.0234       |     0.0175      |     0.0222      |      0.0192     |      0.0192      |
> | FOUL   |   0.1728        |     0.1463      |    0.2902       |     0.1616      |    0.2665       |     0.7684      |    0.5679       |    0.3727       |     0.1668      |       0.1290     |
> | MIUL   |     0.1841      |    0.2069       |    0.1477       |       0.0747    |     0.0719      |     0.0601      |    0.0574       |    0.0683       |      0.0958     |     0.1170       |
>
>
> ***Table 4**: Averaged EO-FPR difference across our unlearning method and the two baselines (optimal: 0)*
> | Method | Epoch 1 | Epoch 10 | Epoch 50 | Epoch 100 | Epoch 200 | Epoch 400 | Epoch 600 | Epoch 800 | Epoch 1000 | Epoch 1500 |
> |:-------|:----------|:----------|:----------|:----------|:----------|:----------|:----------|:----------|:----------|:-----------|
> | **Ours**   |     0.1317      |     0.0462      |     0.0382      |      0.0156     |     0.0257      |      0.0160     |    0.0117       |     0.0207      |        0.0321   |     0.0134       |
> | FOUL   |      0.0752     |     0.0750      |     0.1627      |     0.1071      |     0.3891      |      0.6737     |     0.5946      |    0.3348       |      0.0927     |      0.1227      |
> | MIUL   |      0.1857     |     0.1249      |     0.0318      |   0.0782        |     0.0662      |      0.0557     |     0.1001      |      0.1252     |     0.0941      |       0.0218     |

---

> > ### Comment · Reviewer_nUD2 · 2025-08-01
> >
> > Thank you for your response and the additional experiments in this short time frame.
> >
> > Given that two of your four metrics in the original paper are self-created and lack extensive discussion and validation in the paper (e.g. the shortcomings of SHAP may easily lead to unreliable results), this lowers my confidence in the conclusions from these metrics. Thank you for providing the additional experiments on the privacy side, it is great to see that the method holds up there.
> >
> > No further questions come to mind. All the best!

---

> ### Author Response · Authors · 2025-08-02
>
> Thank you very much for your timely response! We sincerely appreciate the reviewer’s recognition of our additional privacy experiments and are pleased that the strong results further demonstrate the effectiveness of our proposed method.
>
> 1. Regarding the two *“self-created”* evaluation metrics, we respectfully believe there might have been a misunderstanding, and we kindly ask for the opportunity to further their motivation and theoretical grounding.
>
> - The two *“self-created”* evaluation metrics were not arbitrarily defined; rather, they are carefully designed based on core principles of unlearning.
> - Specifically, in an unlearning task, based on the definition, if a feature is successfully unlearned, it should exhibit the property that it no longer contributes to the model’s output (SDI, SRI), and any changes to that feature should have no impact on the model’s predictions (RASI).
>
> 2. Regarding the concern about the “lack of extensive discussion and validation,” **in the original submitted manuscript, we have provided theoretical analysis and proofs supporting these two metrics**.
> - Appendix E provides a proof that our unlearning algorithm causes the Shapley value of the target feature to converge to zero almost surely. The metrics SRI and SDI are designed to directly and empirically measure this provable outcome.
> - Theorem 4.1 and Appendix B show that a successfully unlearned feature should be independent of both other remaining features and the label. RASI aligns with this theoretical foundation, making it a well-justified metric for assessing unlearning effectiveness.
> Thank you for pointing this out and we will make this discussion much clearer in the revised manuscript.
>
> Finally, we would also like to highlight that **our evaluation incorporates all the metrics used in prior studies** and additionally introduces the above theory-backed metrics to provide a more comprehensive and nuanced assessment of unlearning performance.
>
> Thank you again for engaging with our rebuttal. We hope that our clarifications, along with the additional experiments, helps address the remaining concerns regarding the evaluation metrics and further strengthens the overall contribution. As always, we would be happy to respond to any additional questions.
>
> Authors

---

> > ### Comment · Reviewer_nUD2 · 2025-08-03
> >
> > Thank you for adding the clarification. I completely forgot about those proofs amidst the pile of papers to review!
> > Consider these concerns gone and thank you for pushing back - my fault.

---

> > > ### Author Response · Authors · 2025-08-05
> > >
> > > Thank you very much for your thoughtful response. We’re truly pleased to hear that our reply has addressed all your concerns. We deeply appreciate the considerable time and effort you’ve dedicated to reviewing our work. Your constructive and encouraging feedback has been invaluable in helping us improve our work.
> > >
> > > If you feel that our revisions and clarifications have satisfactorily resolved all key issues, we would be most grateful if that could be reflected in your final evaluation/score.
> > >
> > > Thank you again for your generous engagement and invaluable input.

---

> ### Author Response · Authors · 2025-08-09
>
> Dear Reviewer nUD2,
>
> Thank you so much for your positive engagement and for confirming that our rebuttal has addressed all of your concerns. We were very encouraged by your feedback and sincerely appreciate you taking the time to discuss our work.
>
> As the discussion period is closing within 12 hours, we just wanted to send a gentle reminder regarding the final rating. If you have a moment, we would be sincerely grateful if you could update your score in the system to reflect your final assessment.
>
> Thank you again for your valuable time and insightful review.
>
> Authors

---

> > ### Comment · Reviewer_nUD2 · 2025-08-09
> >
> > I waited until the end to see if any other important points would arise in the discussion with other reviewers. The score is updated to "accept" now.

---

### Official Review · Reviewer_U1oC · 2025-06-29

**Clarity:** 2
**Significance:** 2
**Originality:** 2
**Rating:** 4
**Confidence:** 3

**Summary:**

This paper presents a feature unlearning algorithms by first generating corrupted data through feature perturbation based on its possible values uniformly and then fine-tune the model based on the corrupted data. The paper mainly focuses on the its theoretical guarantee of feature correlation removal under the condition of infinite data available.  Experiments compared the proposed approach with two baselines and a tabular data and a image data, but the paper only reported the results of tabular data.

**Questions:**

Could authors simply the proof? The basic idea is to show marginalizing target features contribution can help removing its correlations with other features. The idea should be easily deliverable through less equations. Is it very necessary to show the big chunk of proof in the main paper?

**Ethical Concerns:**

["NO or VERY MINOR ethics concerns only"]

**Final Justification:**

The author provides sufficiently more information about the paper's novelty.

**Limitations:**

No societal impact limitation found in this paper.

**Paper Formatting Concerns:**

No concerns

**Quality:**

2

**Strengths And Weaknesses:**

Strengths:
1. The proof of optimization guarantee looks interesting.

Weaknesses:
1. The proposed method is quite commonly used in practice, spanning from unlearning to model explanations, while it seems not very novel to be published as an independent paper. As early as 2021, a paper [1] with hundreds of citations was published, briefly mentioning the idea.

2. Critiquing existing works but not comparing them. The paper, in its literature review, mentions multiple feature unlearning algorithms, but the authors do not compare the proposed algorithm with those baselines in the paper. Moreover, several well-known baselines were cited but not properly mentioned in the related work section. E.g. [2]

3. The experiment is not complete and also not convincing. The paper mentioned that there are two datasets in the experiments, but it only reported the results for the tabular data. (Yes, I did see there is a six-line description about the CelebA dataset, but the description reads like the authors making it tabular data to use.) In addition, all experimental results are in the form of a "something vs a number of epoch" curve, which is very unconvincing given that multiple metrics are correlated. E.g. EI vs RASI. One can easily reduce EI by increasing the number of epochs. I suggest authors use the proper format to present the paper's results.

4. The literature review is insufficient. Multiple machine unlearning papers mentioned feature unlearning, even though their purpose is often to find optimally searched replacements of target features rather than random sampling. E.g. [3].


[1] Warnecke, Alexander, et al. "Machine unlearning of features and labels." arXiv preprint arXiv:2108.11577 (2021).

[2]Xu, Heng, et al. "Don't Forget Too Much: Towards Machine Unlearning on Feature Level." IEEE Transactions on Dependable and Secure Computing (2024).

[3]Martinsson, John, et al. "Adversarial representation learning for synthetic replacement of private attributes." 2021 IEEE International Conference on Big Data (Big Data). IEEE, 2021.

---

> ### Author Rebuttal · Authors · 2025-07-26
>
> We thank the reviewer for their feedback and for engaging with our work. We have carefully considered the comments and revised our manuscript/clarified our responses accordingly.
>
> **Response to the novelty of our paper**
>
> We would like to respectfully clarify that our method is both **not** "quite commonly used"  and **not** mentioned in reference [1]. Our novelty is summarized in the following three aspects.
> - *Methodological Difference*: Our method performs a **global random permutation** (i.e., shuffling) on the feature to be unlearned. This breaks the feature's statistical correlation with the label and other features. The method in [1] is fundamentally different; it uses influence functions to compute a **local, approximate update** to the model's parameters in response to a specific perturbation (i.e., adding noise or revoking it). **Global shuffling is not discussed in their work.**
> - *Novelty and Literature Context*: To the best of our knowledge, global feature shuffling for unlearning is a new concept in the reviewed existing studies, as supported by comments from **Reviewers ZoQd and nUD2 on our method's novelty**.
> - *Empirical Superiority*: In our original manuscript, **we implemented the method from [1] as a primary baseline (termed FOUL)**. Our experimental results showed that our method consistently and significantly outperforms FOUL across all evaluation metrics.
>
> Hence, our shuffling-based feature unlearning method is a novel contribution to the field, distinct from the influence-function-based approach of reference [1].
>
> [1] Warnecke, Alexander, et al. "Machine unlearning of features and labels." arXiv preprint arXiv:2108.11577 (2021).
>
> **Response to baseline comparison**
>
> Firstly, we would like to clarify that, in our original manuscript, **we identified two key related works in the field of feature unlearning and implemented them to compare with our proposed unlearning algorithm**. The first identified work is the above mentioned FOUL [1] and the other is the work by [2], termed MIUL. The MIUL uses a representation detachment loss defined by mutual information to eliminate specific features. However, this method lacks a theoretical guarantee for the unlearning process. Also, mutual information is simultaneously estimated during unlearning, thereby introducing potential bias. In our original manuscript, our experiments demonstrated the superior performance of our method across all datasets and metrics.
>
> Secondly, we appreciate the reviewer suggesting that some additional baselines were not properly mentioned in the literature review section, such as [3]. In our revised manuscript, we have 1) discussed [3] in the literature review part and 2) implemented [3] as another baseline in addition to FOUL and MIUL.
>
> Our method differs fundamentally from the complex, model-centric adversarial masking used in [3]. Our simpler data permutation (i.e., shuffling) directly breaks statistical correlations and, critically, is supported by theoretical guarantees of achieving statistical independence—a distinct advantage that [3]'s framework lacks.
>
> As an example, the results on the *Credit* dataset below confirm our method's effectiveness against this newly added baseline. The complete experiments and results have been updated in the revised manuscript.
>
> | Evaluation Method | Algorithm | Epoch 1 | Epoch 10 | Epoch 50 | Epoch 100 | Epoch 200 | Epoch 400 | Epoch 600 | Epoch 800 | Epoch 1000 | Epoch 1500 |
> | :--- | :--- | :--- | :--- | :--- | :--- | :--- | :--- | :--- | :--- | :--- | :--- |
> | **$TRI$** |[3] | 102% ± 0.4% | 102% ± 0.6% | 102% ± 0.3% | 102% ± 0.5% | 102% ± 0.7% | 102% ± 0.5% | 101% ± 0.5% | 102 ± 0.5% | 102% ± 0.4% | 102% ± 0.3% |
> | | **Ours** | **100% ± 1.0%** | **99% ± 0.8%** | **99% ± 0.8%** | **100% ± 0.7%** | **100% ± 0.6%** | **100% ± 0.4%** | **100% ± 0.5%** | **100% ± 0.5%** | **100% ± 0.6%** | **99% ± 0.7%** |
> | **$EI$** | [3] | 1.7 | 1.7 | 1.5 | 1.4 | 1.1 | 0.8 | 0.7 | 0.5 | 0.5 | 0.3 |
> | | **Ours** | **2431** | **272** | **47** | **25** | **11** | **5.7** | **3.8** | **2.7** | **1.7** | **1.2** |
> | **$RASI$** | [3] | 83% ± 0.4% | 83% ± 0.4% | 83% ± 0.4% | 83% ± 0.6% | 83% ± 0.4% | 83% ± 0.3% | 83% ± 0.4% | 83% ± 0.6% | 83% ± 0.5% | 83% ± 0.6% |
> | | **Ours** | **88% ± 1.1%** | **93% ± 0.5%** | **94% ± 0.5%** | **95% ± 0.3%** | **96% ± 0.3%** | **97% ± 0.4%** | **98% ± 0.2%** | **99% ± 0.2%** | **99% ± 0.1%** | **99% ± 0.2%** |
> | **$SRI$** | [3] | 103% ± 2.9% | 103% ± 3.2% | 103% ± 3.5% | 102% ± 4.3% | 103% ± 2.8% | 102% ± 3.0% | 101% ± 2.4% | 100% ± 3.7% | 102% ± 4.6% | 99% ± 3.5% |
> | | **Ours** | **59% ± 10%** | **25% ± 5.8%** | **20% ± 2.8%** | **17% ± 3.5%** | **13% ± 1.6%** | **9.0% ± 3.4%** | **7.0% ± 1.1%** | **4.0% ± 1.4%** | **4.0% ± 1.0%** | **0.3% ± 1.6%** |
> | **$SDI$** | [3] | 13% ± 0.4% | 13% ± 0.4% | 13% ± 0.5% | 13% ± 0.6% | 13% ± 0.4% | 13% ± 0.4% | 13% ± 0.3% | 13% ± 0.5% | 13% ± 0.6% | 13% ± 0.5% |
> | | **Ours** | **8.0% ± 1.3%** | **3.0% ± 0.8%** | **3.0% ± 0.4%** | **2.0% ± 0.5%** | **2.0% ± 0.2%** | **1.0% ± 0.4%** | **1.0% ± 0.1%** | **0.1% ± 0.2%** | **0% ± 0.1%** | **0% ± 0.2%** |
>
> [2] Tao Guo, Song Guo, Jiewei Zhang, Wenchao Xu, and Junxiao Wang. Efficient attribute
> unlearning: Towards selective removal of input attributes from feature representations, 2022.
>
> [3]Xu, Heng, et al. "Don't Forget Too Much: Towards Machine Unlearning on Feature Level." IEEE Transactions on Dependable and Secure Computing (2024).
>
> **Response to used datasets in experiments**
>
> Thank you for the opportunity to clarify our experiments. **We would like to highlight that the CelebA image dataset was not converted into a tabular format. Our unlearning method was applied directly to the visual features within the images themselves.**
>
> As detailed in Section 5.2 of the original manuscript, we addressed the challenge of implicit image features by shuffling specific image regions corresponding to semantic attributes. For the used CelebA image dataset, we unlearned features like "nose" and "eyes" by shuffling the corresponding image patches across the training set. Appendix G (in the original manuscript) provides a clear visual example of this process.
>
> These image-based experiments, with full results for single and multi-feature scenarios in Figure 3 and Appendix N (in the original manuscript), complement our extensive evaluations on **six diverse tabular datasets**. **Our experiments encompass seven datasets in total (six tabular and one image datasets), not just two**. Various features has been unlearned within each used dataset to validate the effectiveness of our method. We trust this confirms the comprehensive nature of our experimental validation.
>
> **Response to evaluation metrics**
>
> We chose the "metric vs. number of epochs" curves to illustrate the **convergence dynamics** of the unlearning process, which we believe is **a crucial aspect of evaluation**. These plots show not just the final outcome, but also how efficiently each method reaches an effective state. For example, our results demonstrate that our unlearning method consistently achieves more effective unlearning (e.g., high RASI and TRI near 1.0) *within a relatively small number of epochs*. At this point of convergence, the efficiency (EI) remains very high, **showcasing that our method does not require sacrificing efficiency for effectiveness**.
>
> Regarding the EI vs. RASI trade-off, while it is true that running more epochs generally improves unlearning (higher RASI) at the cost of efficiency (lower EI), our experiments show that **our method reaches a near-perfect RASI long before the EI becomes impractically low**. This rapid convergence is a key advantage over baselines, which may converge slowly or fail to reach an acceptable level of unlearning regardless of the number of epochs.
>
> To make the results more direct and easier to compare, in the revised manuscript, we have adopted the format used in our multi-feature unlearning analysis (Tables 7, 8, and 9) and added summary tables for the single-feature experiments. These tables now report the values for all key metrics (TRI, EI, RASI, SRI, SDI) at a specific epoch count where our method has clearly converged, providing a clear, at-a-glance comparison that complements the dynamic curves.
>
> **Response to insufficient literature review**
>
> In the revised manuscript, we have included the reference [4] (that the reviewer mentioned) into our literature review as follows. The work [4] proposes a data privatization technique, not exactly a model unlearning method, that uses an adversarial filter-generator architecture to first remove a sensitive attribute and then synthetically replace it with a new, randomly sampled value. Their goal is to create a transformed, privacy-preserving dataset, which fundamentally differs from machine unlearning's objective of modifying a pre-trained model to make it forget learned information. Consequently, while our methods directly alter an existing model's parameters to achieve the unlearning of features, their technique serves as a data pre-processing step to anonymize a dataset before its use.
>
> [4]Martinsson, John, et al. "Adversarial representation learning for synthetic replacement of private attributes." 2021 IEEE International Conference on Big Data (Big Data). IEEE, 2021.
>
>
> **Response to proof simplification**
>
> In the revised manuscript, we have streamlined the proof presentation. In addition, we would like to highlight that, though the idea of shuffling seems to be intuitive, a rigorous proof is necessary to formally guarantee that our method is effective. Our proof first establishes that shuffling creates statistical independence and then connects this property to the final model, showing it is $(\epsilon,\delta)$-close to a retrained one.
>
> In the revised manuscript, we have included concise proof sketch in the main paper to improve readability and moved the detailed, complete proof to the appendix.

---

> > ### Comment · Reviewer_U1oC · 2025-08-05
> >
> > Thanks for putting my previous suggestions into your promised revisions.
> >
> > One more question regarding the practical applicability of the proposed method:
> >
> > What information does the method require at unlearning time? In practice, data points are often no longer available due to data retention regulations. What statistics or representations are needed for the method to work without access to the original data?
> >
> > In the image unlearning experiments, the authors mention removing features like the nose or eyes. However, such features are part of the original images and cannot typically be stored or reused due to privacy constraints. Similarly, for tabular data, non-categorical (continuous) features pose the same challenge—retaining them may violate data retention policies. How does the proposed method handle these limitations in a realistic setting?

---

> > > ### Author Response · Authors · 2025-08-06
> > >
> > > Thank you for your continued engagement and valuable feedback.
> > >
> > > First, we would like to take this opportunity to clarify that our primary objective is to develop a method that **rigorously removes a model's dependence on specified features**, thereby achieving a state that is **provably equivalent to having retrained the model from scratch** by accessing the *original dataset*.
> > > - The **core challenge  in machine unlearning is** not merely circumventing access to the original training data, but ensuring the **complete and verifiable removal of information**. Our work provides theoretical guarantees that, given access to the training dataset, our unlearning method yields a model that is $(\epsilon-\delta)$-close to the model retrained from scratch. Full data access lets us reshuffle the target feature—erasing its predictive power while leaving every other statistical tie untouched.
> > > - **This assumption of data access is not a matter of convenience but is widely recognized in the literature as a prerequisite for achieving certified or "exact" unlearning**. For instance, the highly-cited study [1] establishes that for an unlearning algorithm to produce a model provably $(\epsilon-\delta)$-close to a retrained model, it must operate on the original training dataset (see Equations (6) and (7), Page 8 of [1]). Consequently, **methods without access to the training data can, at best, achieve *”approximate”* unlearning**, which may not satisfy the stringent requirements for auditable and faithful data removal.
> > >
> > > [1] Nguyen, Thanh Tam, et al. "A survey of machine unlearning." arXiv preprint arXiv:2209.02299 (2022).
> > >
> > > Second, we acknowledge that in certain operational contexts, persistent access to the original training data may be restricted or limited. Our framework can be adapted for such scenarios, although this would likely come at the cost of the rigorous theoretical guarantees established in our work. For instance, one potential extension could leverage privacy-preserving statistical summaries (e.g., marginals or conditional histograms) captured before deletion requests to support "approximate” feature unlearning via synthetic data regeneration and pseudo-shuffling.
> > >
> > > However, and more importantly, **our method is directly applicable to the most common real-world paradigm for handling data deletion: supervised access**. Regulatory frameworks like the GDPR legally obligate data controllers (e.g., Google, Microsoft) to fulfill data deletion requests. Compliance, overseen by Data Protection Authorities (DPAs) in each European Union member state, is managed through internal systems like Google's Data Management and Microsoft's Compliance Manager. **These systems necessarily interact with the original data under strict supervision to ensure deletion is performed correctly and verifiably.** Our unlearning approach can be well-suited to these supervised environments, bridging the gap between theoretical guarantees and practical, regulatorily-compliant implementation.
> > >
> > > We hope our response clarifies our design choices and demonstrates the alignment of our work with both foundational theory and real-world practice. We will incorporate this discussion in our revised manuscript.
> > >
> > > We sincerely appreciate your time and thoughtful review, and welcome any further specific comments or questions to help us improve this paper.

---

> > > > ### Author Response · Authors · 2025-08-08
> > > >
> > > > Dear Reviewer U1oC:
> > > >
> > > > As the rebuttal period will close in fewer than 48 hours, we would like to confirm that our replies have fully addressed your concerns.
> > > >
> > > > In our previous responses we have:
> > > >
> > > > - **Clarified our focus on exact unlearning with theoretical guarantee:**  achieving provable equivalence to retraining requires full access to the original dataset, as established in the cited literature.
> > > >
> > > >
> > > > - **Outlined the approach and limits of restricted access to the data:** -  when data access is restricted, one can resort to privacy-preserving summaries or synthetic data, but these approaches sacrifice the formal guarantees and lie outside the scope of our work.
> > > >
> > > >
> > > > - **Shown real-world relevance of our approach:**  the proposed method aligns with supervised-access workflows mandated by regulations such as the GDPR, where controllers (e.g., Google, Microsoft) operate on original data under regulator oversight to verify deletion.
> > > >
> > > > If any issues remain unresolved, please let us know—we would be happy to clarify them promptly. If you have no further specific concerns, we would be sincerely grateful if you could consider updating your score accordingly.
> > > >
> > > > Authors

---

> ### Author Response · Authors · 2025-08-09
>
> Dear Reviewer U1oC,
>
> With the rebuttal period closing in less than 12 hours, we would like to respectfully follow up on our previous response. We hope we have been able to fully address your concerns and wanted to briefly summarize our clarifications.
>
> In response to your review, we have:
> - Clarified our focus on exact unlearning: We explained that our goal of providing theoretical guarantees for unlearning necessitates access to the original dataset, a standard requirement in this specific subfield.
> - Discussed the scope regarding restricted data access: We acknowledged that approaches for restricted-access settings exist (e.g., using synthetic data or privacy-preserving summaries), but they cannot offer the formal equivalence guarantees that are the core of our work.
> - Demonstrated real-world relevance: We highlighted that our model can be adopted under supervised access, thereby directly aligned with regulatory frameworks like the GDPR, where data controllers (such as Google and Microsoft) must verifiably prove deletion on the original data.
>
> We would be very grateful for the opportunity to address any final questions you might have. If our rebuttal has clarified the points of concern, we would sincerely appreciate it if you would consider updating your score accordingly.
>
> Thank you for your time and valuable feedback.
>
> Authors

---

> > ### Comment · Reviewer_U1oC · 2025-08-09
> >
> > I have no further questions. Thanks for the detailed response.

---

### Official Review · Reviewer_ZoQd · 2025-06-30

**Clarity:** 2
**Significance:** 2
**Originality:** 2
**Rating:** 4
**Confidence:** 2

**Summary:**

This paper presents a feature unlearning method that removes the influence of specific features from trained models. Unlike traditional machine unlearning methods focused on data instance removal, the work introduces a shuffling-based feature unlearning method, where target feature values are randomly shuffled in the training data, followed by fine-tuning. The paper provides theoretical guarantees that this process removes statistical dependency between the shuffled features. Theoretical validation is supported through mutual information analysis and Shapley value equivalence. The authors propose clear metrics (e.g., TRI, EI, RASI, SRI, SDI) and validate their method across tabular and image datasets, showing that their method achieves comparable performance to retraining from scratch while being significantly more efficient.

**Questions:**

Please see Weaknesses.

**Ethical Concerns:**

["NO or VERY MINOR ethics concerns only"]

**Final Justification:**

The rebuttal does a good job in explaining the theoretical context and assumptions, which I appreciate. Although not at the level I would ideally like it to be, the validation is also better now. Overall, a good paper that ia ready to be published with the additional information in the rebuttal.

**Limitations:**

Yes.

**Quality:**

3

**Strengths And Weaknesses:**

Strengths:

* The paper presents a simple yet rigorously supported method with clear mathematical foundations and convergence proofs. The methodology is thoughtfully constructed and the empirical evaluations are comprehensive.

* While prior work has studied instance unlearning or model detachment, the idea of using random feature shuffling followed by fine-tuning as a strategy for feature-level unlearning seems novel.

* The empirical results across both tabular and image datasets show that the proposed method removes influence of designated features and preserves the information content of remaining features.

Weaknesses:

* The proposed method requires access to the full training dataset. This dependency may be impractical in scenarios where data retention is restricted post-training or where training is performed on large-scale datasets that are costly to store or access.

* The theoretical foundations rely on assumptions that may not hold in real-world settings, especially for datasets exhibiting temporal drift or non-i.i.d. behavior.

* While FOUL and MIUL are evaluated, the empirical study could be strengthened by including additional baselines—particularly those involving adversarial feature disentanglement or post-hoc pruning strategies.

* The motivation includes ethical concerns related to sensitive attributes in image domains, yet no experiments explicitly explore the removal of attributes such as race or gender which would be a more meaningful case for machine unlearning.

* To further demonstrate robustness and effectiveness in real-world scenarios, experiments on more complex visual features for image classification are needed. Currently, the image task focuses on removing relatively simple features (e.g., nose and eyes), with classification tasks including gender, big nose, pointy nose, eyeglasses, and narrow eyes. Expanding the target attributes for removal and classification would better showcase the method’s generalizability.

* Qualitative examples could improve clarity. Although Appendix G includes examples of processed images with shuffled visual features, it only provides a single example. Including more diverse visual results would help illustrate the method’s behavior and impact.

---

> ### Author Rebuttal · Authors · 2025-07-30
>
> We thank the reviewer for their feedback and for engaging with our work. We have carefully considered the comments and revised our manuscript/clarified our responses accordingly.
>
> ***Response to the assumption of access to the full training dataset***
>
> Our approach deliberately assumes access to the full training dataset, since the core mandate of machine unlearning is to eradicate the model’s dependence on specified data—not merely to sidestep access to the original training set. Methods that impose a rigid “no-access-to-data” constraint fundamentally compromise the integrity of unlearning by prioritising convenience over correctness, thereby failing to achieve robust, auditable, and faithful unlearning. Such a trade-off thus results in diluted guarantees and unverifiable unlearning outcomes.
>
> Moreover, as the aim of our method is to re-establish the original empirical joint distribution while selectively breaking the dependence on the target feature, it is only achievable by reshuffling the feature across the entire training set (as shown in our theoretical proof), ensuring that all statistical relationships—especially between correlated variables—are properly preserved or neutralised. Without full data access, such reshuffling would introduce unintended distribution shifts or artifacts, thereby compromising the validity of unlearning.
>
> We acknowledge that in certain regulatory or operational contexts, full access to the original training data may not be possible at unlearning time. In such cases, our method may not be directly applicable. However, we view this limitation not as a flaw, but as a deliberate design choice to prioritise unlearning fidelity and theoretical guarantees. Organisations can retain light-weight, privacy-preserving statistics (e.g., marginal feature distributions or conditional histograms) computed before deletion requests. These can support approximate feature removal via pseudo-shuffling or data regeneration, though the resulting guarantees will necessarily be weaker than our full-data method.
>
> ***Response to the assumption of temporal shift or non-i.i.d. behavior***
>
> We adopt the i.i.d. and stationarity assumptions for three main reasons:
> - The assumptions of i.i.d. samples and stationarity serve as a standard theoretical simplification that ensures mathematical tractability and allows us to build a rigorous understanding of model behaviour *under controlled conditions*.
>
> - Moreover, our empirical evaluations demonstrate strong performance of our method across all tested datasets, indicating empirical robustness that extends beyond the idealised theoretical assumptions. Such empirical success reveal the capability of our method in handling real-world datasets which may not fully align with our assumptions and have substantial internal dependencies and correlations.
>
> - These assumptions are standard throughout the machine learning and unlearning literature [1][2][3], and form the foundation of nearly all theoretical analyses of unlearning–retraining equivalence, e.g., SISA, FOUL, MIUL, where the latter two have been implemented as baselines in our experiments. Such frameworks consistently adopt empirical risk minimisation under the premise of a stable, stationary data-generating process.
>
> In the revised manuscript, we have made this discussion clearer and explicitly distinguish between theoretical guarantees and practical robustness.
>
> [1] Warnecke, Alexander, et al. "Machine unlearning of features and labels." arXiv preprint arXiv:2108.11577 (2021).
>
> [2] Tao Guo, Song Guo, Jiewei Zhang, Wenchao Xu, and Junxiao Wang. Efficient attribute unlearning: Towards selective removal of input attributes from feature representations, 2022.
>
> [3] Xu, Heng, et al. "Don't Forget Too Much: Towards Machine Unlearning on Feature Level." IEEE Transactions on Dependable and Secure Computing (2024).
>
> ***Response to additional baselines that involving adversarial feature disentanglement or post-hoc pruning***
>
> First, we would like to clarify that **FOUL**, the first baseline we evaluated, **is itself an adversarial method**. Specifically, FOUL trains a feature discriminator in an adversarial setting to ensure that the representation does not encode information about the feature to be unlearned. This aligns closely with the family of adversarial feature disentanglement techniques the reviewer referred to.
>
> Secondly, we include **additional baselines** that employ a **model-centric adversarial masking** strategy [3], in addition to two already-implemented baselines, i.e., FOUL and MIUL. Notably, our approach is fundamentally different from the strategy used in [3]. Instead of introducing complex architectural modifications, our method employs a simple yet effective data permutation (i.e., shuffling) technique that directly disrupts statistical dependencies. More importantly, our approach is backed by **theoretical guarantees** of achieving statistical independence, which distinguishes it from methods like [3] that lack such formal assurances.
>
> As an example, the results on the Credit dataset below confirm our method's effectiveness against this newly added baseline.
> |Eva.Me.|Algo.|1|10|50|100|200|400|600|800|1000|1500|
> |:---|:---|:---|:---|:---|:---|:---|:---|:---|:---|:---|:---|
> |**$TRI$**|[3]|102% ± 0.4%|102%±0.6%|102%±0.3%|102%±0.5%|102%±0.7%|102%±0.5%|101%±0.5%|102±0.5%|102%±0.4%|102%±0.3%|
> | | Ours |**100%±1.0%**|**99%±0.8%**|**99% ± 0.8%**| **100% ± 0.7%** | **100% ± 0.6%** | **100% ± 0.4%** | **100% ± 0.5%** | **100% ± 0.5%** | **100% ± 0.6%** | **99% ± 0.7%** |
> | **$EI$** | [3] | 1.7 | 1.7 | 1.5 | 1.4 | 1.1 | 0.8 | 0.7 | 0.5 | 0.5 | 0.3 |
> | | Ours | **2431** | **272** | **47** | **25** | **11** | **5.7** | **3.8** | **2.7** | **1.7** | **1.2** |
> | **$RASI$** | [3] | 83% ± 0.4% | 83% ± 0.4% | 83% ± 0.4% | 83% ± 0.6% | 83% ± 0.4% | 83% ± 0.3% | 83% ± 0.4% | 83% ± 0.6% | 83% ± 0.5% | 83% ± 0.6% |
> | | Ours | 88% ± 1.1% | **93% ± 0.5%** | **94% ± 0.5%** | **95% ± 0.3%** | **96% ± 0.3%** | **97% ± 0.4%** | **98% ± 0.2%** | **99% ± 0.2%** | **99% ± 0.1%** | **99% ± 0.2%** |
> | **$SRI$** | [3] | 103% ± 2.9% | 103% ± 3.2% | 103% ± 3.5% | 102% ± 4.3% | 103% ± 2.8% | 102% ± 3.0% | 101% ± 2.4% | 100% ± 3.7% | 102% ± 4.6% | 99% ± 3.5% |
> | | Ours | 59% ± 10% | **25% ± 5.8%** | **20% ± 2.8%** | **17% ± 3.5%** | **13% ± 1.6%** | **9.0% ± 3.4%** | **7.0% ± 1.1%** | **4.0% ± 1.4%** | **4.0% ± 1.0%** | **0.3% ± 1.6%** |
> | **$SDI$** | [3] | 13% ± 0.4% | 13% ± 0.4% | 13% ± 0.5% | 13% ± 0.6% | 13% ± 0.4% | 13% ± 0.4% | 13% ± 0.3% | 13% ± 0.5% | 13% ± 0.6% | 13% ± 0.5% |
> | | Ours | 8.0% ± 1.3% | **3.0% ± 0.8%** | **3.0% ± 0.4%** | **2.0% ± 0.5%** | **2.0% ± 0.2%** | **1.0% ± 0.4%** | **1.0% ± 0.1%** | **0.1% ± 0.2%** | **0% ± 0.1%** | **0% ± 0.2%** |
>
> ***Response to motivation related to unlearning sensitive attributes***
>
> Our method is capable of unlearning **any individual feature or combination of features**, **including sensitive attributes, like gender or race** (referred to Definition 3.1 and Theory 4.4 in the original manuscript). This flexibility is a direct consequence of both our *theoretical guarantees* and *empirical design*. Specifically, we provide formal proof that our approach removes statistical dependence between the unlearned feature and the model’s representation or output—regardless of the semantic nature of the feature itself.
>
> To support this claim, we also conducted extensive evaluations across a wide range of **tabular and visual datasets**. In particular, for the visual datasets, we have targeted diverse features such as **nose shape, eye size, and eyeglasses**. These features vary substantially across individuals and may capture latent demographic cues, including those related to identity and fairness.
>
> Further, we highlight our results on the **CALI** dataset, where we successfully unlearned the **DebtRatio feature** (see Appendix L, table 6, page 31 in the original manuscript)—a **sensitive financial indicator** tied to personal debt levels. The ability of our method to remove this feature while preserving model utility further underscores its effectiveness for privacy- and fairness-sensitive applications.
>
> In the revised manuscript, we have clarified these points and expanded our discussion on how our approach applies to unlearning sensitive features such as race, gender, and socioeconomic attributes—areas that are closely aligned with the core motivation of our work.
>
> ***Response to experiments on more complex visual features***
>
> The authors are not entirely certain what is meant by *“more complex visual features,”* but we attempt to address the question based on our interpretation. *In raw, pixel-based human facial images*, *features such as the eyes, eyebrows, nose, mouth, and ears are semantically identical*. Our method has already been evaluated on key features like the eyes, nose, and their combination (eyes + nose) across various classification tasks. **Adding additional facial attributes of a similar nature does not introduce further methodological complexity** for our approach, which directly processes raw image inputs without relying on specialised preprocessing. Moreover, identifying specific facial regions such as the nose or eyes or more ambiguous features is not the main focus of our study.
>
> If we have misunderstood the reviewer’s request, please let us know—we would be happy to provide additional clarification or experiments as needed.
>
> ***Response to more qualitative examples***
>
> In Appendix G of the original manuscript, we provided an image example illustrating the unlearning task for the nose feature to demonstrate the core idea behind our process. In the revised manuscript, we have expanded Appendix G to include a broader set of qualitative examples, covering all the test cases conducted on the image dataset. The enriched results now include diverse identities and multiple feature removal scenarios to provide a more comprehensive visual summary of the method’s effect.

---

> > ### Author Response · Authors · 2025-08-06
> >
> > As the rebuttal period is ending in less than three days, we kindly wanted to check whether our response has fully addressed your concerns. If there are any remaining questions or clarifications needed, we’d be more than happy to respond promptly.
> >
> > If your concerns have been resolved, we would greatly appreciate it if you could consider adjusting your score accordingly.

---

> > > ### Author Response · Authors · 2025-08-07
> > >
> > > As the rebuttal period ends in less than 48 hours, we wanted to kindly confirm whether our response has fully addressed your concerns. In our rebuttal, we believe we have carefully responded to each of the points you raised:
> > > - We have further clarified the context for our theoretical assumptions, demonstrating their practicality and applicability in real-world scenarios.
> > > - We have expanded our experiments to include a new adversarial baseline, providing a more robust and convincing empirical evaluation.
> > > - We have elaborated in detail on how our theoretical guarantees translate to the successful unlearning of both sensitive attributes and complex visual features.
> > > - We have improved the paper's clarity by incorporating more qualitative examples to better illustrate our method's capabilities.
> > >
> > > We believe our response has fully addressed your concerns and has substantially improved the clarity and overall quality of the paper. If you have no further specific concerns, we would be sincerely grateful if you could consider updating your score accordingly.
> > >
> > > Authors

---

> > > > ### Comment · Reviewer_ZoQd · 2025-08-07
> > > >
> > > > Dear Authors,
> > > >
> > > > Apologies that I could not respond earlier and thanks for the rebuttal. Yes, this rebuttal adressed most of my comments and I am happy to raise my score.
> > > >
> > > > Best.

---

> > > > > ### Author Response · Authors · 2025-08-08
> > > > >
> > > > > Thank you very much for your kind response and for taking the time to consider our rebuttal. We’re truly grateful that our clarifications were helpful and that you are willing to raise your score. Thank you again for your thoughtful feedback and support!!!
> > > > >
> > > > > Authors

---

### Official Review · Reviewer_wgRy · 2025-07-03

**Clarity:** 2
**Significance:** 3
**Originality:** 2
**Rating:** 4
**Confidence:** 3

**Summary:**

The paper proposes a two-step method approach for selectively forgetting features: first shuffles the target feature(s) in the original training data, and then fine-tune the original model on this shuffled dataset. They provide a theoretical guarantee of this proposed method for effective feature unlearning, and also empirically show the proposed method is effective and efficient across experiments across both tabular and image datasets.

**Questions:**

1. The paper includes unlearning experiments on different types of features, e.g., features with high Shapley importance or those strongly correlated with other inputs. Have you observed any patterns or differences in how your method (or the baselines) perform across these cases? Are there particular feature types that are easier or harder to unlearn?

**Ethical Concerns:**

["NO or VERY MINOR ethics concerns only"]

**Final Justification:**

Most of my concerns have been adequately addressed in the rebuttal. I appreciate the authors’ clarifications and additional results. However, I share the concern raised by Reviewer U1oC regarding the novelty of the proposed method. The idea of random shuffling has been commonly used in prior unlearning work (e.g., [1] for example-level unlearning), though not explicitly in the context of feature unlearning. That said, I acknowledge the empirical effectiveness of the proposed method in the feature unlearning setting. This reduces the weight I assign to the novelty concern in my evaluation.

No major issues remain unresolved from my side, and I found the authors’ responses thoughtful and helpful, so I decide to maintain my initial rating.

[1] Fan, Chongyu, et al. "Salun: Empowering machine unlearning via gradient-based weight saliency in both image classification and generation."

**Limitations:**

See Weaknesses section.

**Quality:**

3

**Strengths And Weaknesses:**

Strengths:
1. The proposed algorithm is simple and practical to implement. It requires only permuting a selected feature and performing standard fine-tuning.
2. The paper offers a sound theoretical analysis of the unlearning process.
3. The empirical evaluation is fairly comprehensive, using multiple complementary metrics to assess both unlearning effectiveness and computational efficiency. The experiments also cover a diverse range of datasets and architectures.

Weaknesses:
1. The writing and overall presentation could be improved. Some notations, particularly in the formal definitions and theoretical sections, are confusing or insufficiently explained.

---

> ### Author Rebuttal · Authors · 2025-07-30
>
> We thank the reviewer for their feedback and for engaging with our work. We have carefully considered the comments and revised our manuscript/clarified our responses accordingly.
>
> ***Response to writing and overall presentation***
>
> Thank you for your feedback regarding the clarity of writing and notation. We appreciate the reviewer pointing this out, and we have taken steps to improve the overall presentation in the revised manuscript. Specifically, we have revised and clarified the notations in the formal definitions and theoretical sections, particularly in places where some symbols may have benefited from additional explanation. To further aid readability, we have included a dedicated table of notations in Appendix, which consolidates the definitions of all symbols, parameters, and variables used throughout the paper. This appendix also includes a brief explanation of the key mathematical tools we rely on (e.g., total variation distance and conditional independence) for readers who may not be familiar with them.
>
> We hope these revisions make the theoretical sections more accessible and precise, and we welcome any further suggestions to improve clarity.
>
> ***Response to the observations on patterns of unlearning various features***
>
> Thank you for this thoughtful question. We do observe some meaningful patterns in how different types of features affect unlearning performance, although these were only briefly mentioned in the original experimental section. To address this, we will expand the discussion in the revised manuscript to provide a more detailed analysis.
>
> For single-feature unlearning, our method shows consistently strong performance, regardless of the importance or influence of the target feature. As shown in Figure 2 and Appendix F, even for features with high SHAP values, our approach achieves near-complete unlearning within approximately 150 epochs. This stability stems from the fact that our method explicitly removes statistical dependence by reshuffling the target feature across the entire training set, effectively breaking both direct and indirect correlations. The process does not rely on learning a complex adversarial signal, making it less sensitive to the feature's original importance.
> In contrast, baselines such as FOUL tend to perform better on less important features and struggle more with high-impact ones. This is likely because FOUL relies on an adversarial feature discriminator, which may face difficulty suppressing deeply entangled or high-importance signals within the model representation.
>
> That said, we did observe that unlearning multiple features simultaneously is more challenging for all methods, including ours. These cases generally require longer training and exhibit slightly lower performance, likely due to the increased complexity of disentangling joint dependencies among multiple correlated features.
>
> We will clarify these observations in the revised manuscript and expand our discussion in Appendix F to include a more detailed breakdown of performance trends across different feature types.

---

> > ### Comment · Reviewer_wgRy · 2025-08-04
> >
> > Thank you for your response. My concerns have been addressed, and I have no further questions. Good luck!

---

> > > ### Author Response · Authors · 2025-08-05
> > >
> > > We thank the reviewer for the kind follow-up and are pleased to hear that our response has addressed all your concerns. We will ensure your suggestions are fully incorporated into the revised manuscript. Your guidance has been invaluable in helping strengthen our work.
> > >
> > > If you feel that the revisions and clarifications have resolved all key issues, we would be sincerely grateful if this could be reflected in your final evaluation/score.
> > >
> > > Thank you again for your thoughtful feedback and generous engagement.

---

### Note · Authors · 2025-08-13

We sincerely thank all four reviewers for their thoughtful feedback and engagement, and the Area Chair for coordinating a constructive discussion.

We are encouraged to see that our rebuttal and responses were positively received, and **all four reviewers explicitly stated that their concerns have been fully addressed in our rebuttal and discussion**.

Across the reviews, there is clear and consistent agreement on several key strengths of our paper:

- **Novelty, Simplicity, and Practicality:** Reviewers nUD2, ZoQd, and wgRy all highlighted that our method is simple, elegant, novel, and practical to implement.
- **Theoretical Soundness:** Our work is recognized for its “sound theoretical analysis” (wgRy) , “clear mathematical foundations and convergence proofs” (ZoQd), and “simple and elegant method with theoretical backing” (nUD2). Reviewer U1oC noted our proof was "interesting."
- **Comprehensive Evaluation:** Our experiments were praised as “extensive” (nUD2) and "fairly comprehensive" (wgRy), effectively validated across multiple metrics, diverse datasets (including both tabular and image data), and different model architectures. The reviewer ZoQd also noted that “empirical results across both tabular and image datasets show that the proposed method removes influence of designated features and preserves the information content of remaining features.”

We also wish to respectfully highlight that machine unlearning is a rapidly emerging research direction of increasing importance, especially in light of growing attention to data privacy and responsible AI. We are encouraged by the **thoughtful recognition of this topic’s significance by the reviewers** and we believe that our work contributes meaningfully to its development.

We are committed to thoughtfully integrating all suggestions (including the additional evaluation metrics, discussion, and experiments ) to ensure the final version reflects the highest standards of clarity, rigor, and impact that this important topic merits.

We hope the consensus reached during the discussion, coupled with the alignment of our work with the community’s values on impactful, technically sound, and broadly useful contributions, will support a positive recommendation.

Once again, thank you for your time, thoughtful consideration, and dedication to fostering high-quality research.

Authors

---

### Decision · Program_Chairs · 2025-09-17

**Decision:**

Accept (poster)

**Comment:**

In the inital reviews, this work received mixed reviews. After rebuttal, this work receives all positive ratings. The rebuttal adequately addressed the concerns of the reviewers. The meta reviewer agrees with the reviewers' recommendation.